# CLIP21: ERROR FEEDBACK FOR GRADIENT CLIPPING

## ABSTRACT

Motivated by the increasing importance of deep neural network training, we study distributed gradient methods with *gradient clipping*, i.e., clipping applied to the gradients computed from local information at the nodes. While gradient clipping enforces the convergence of gradient-based methods that minimize rapidly growing functions, it also induces bias which causes serious convergence issues specific to the distributed setting. Inspired by recent progress in the error-feedback literature which is focused on taming the bias/error introduced by communication compression operators such as Top-$k$ (Richtárik et al., 2021), and mathematical similarities between the clipping operator and contractive compression operators, we design Clip21 – the first provably effective and practically useful error feedback mechanism for distributed methods with gradient clipping. We prove that our method converges at the same $\mathcal{O}(1/K)$ rate as distributed gradient descent in the smooth nonconvex regime, which improves the previous best $\mathcal{O}(1/\sqrt{K})$ rate which was obtained under significantly stronger assumptions. Our method converges significantly faster in practice than competing methods.

## 1 INTRODUCTION

Gradient clipping is a popular and versatile tool used in several areas of machine learning. For example, it is employed to i) enforce bounded $\ell_2$ sensitivity in order to obtain formal differentially privacy guarantees in gradient-based optimization methods (Abadi et al., 2016; Chen et al., 2020), ii) tame the exploding gradient problem in deep learning (Pascanu et al., 2013; Zhang et al., 2020a), iii) stabilize convergence of SGD in the heavy-tailed noise regime Nazin et al. (2019); Gorbunov et al. (2020a), and iv) design provably Byzantine-robust gradient aggregators (Karimireddy et al., 2021).

Our work is motivated by the increasing popularity and importance of training deep neural networks that are prone to exploding gradient problems (Pascanu et al., 2013; Goodfellow et al., 2016; Mai & Johansson, 2021). In particular, we wish to solve the optimization problem

$$\min_{x \in \mathbb{R}^d} \left[ f(x) := \frac{1}{n} \sum_{i=1}^{n} f_i(x) \right], \tag{1}$$

where $n$ is the number of clients and $f_i$ is the loss of a model parameterized by vector $x \in \mathbb{R}^d$ over all private data $\mathcal{D}_i$ owned by client $i \in [n] := \{1, \ldots, n\}$. We assume each $f_i$ has $L_i$-Lipschitz gradient, i.e.,

$$\|\nabla f_i(x) - \nabla f_i(y)\| \le L_i \|x - y\| \tag{2}$$

for all $x, y \in \mathbb{R}^d$, where $\|x\| := \langle x, x \rangle^{1/2}$ and $\langle x, y \rangle := \sum_{i=1}^{d} x_i y_i$ is the standard Euclidean inner product. This implies that $f$ has $L$-Lipschitz gradient, i.e.,

$$\|\nabla f(x) - \nabla f(y)\| \le L \|x - y\| \tag{3}$$

for all $x, y \in \mathbb{R}^d$, where $L$ satisfies $L \le \frac{1}{n} \sum_{i=1}^{n} L_i$. Let $L_{\max} := \max_i L_i$. Moreover, we assume $f$ to be lower bounded by some $f_{\inf} \in \mathbb{R}$. Note that we do not require any convexity assumptions.

### 1.1 OPTIMIZATION WITH GRADIENT CLIPPING

Perhaps the simplest algorithm for solving Equation 1 in this regime is Clip-GD (Liu et al., 2022; Zhang et al., 2020a), performing the iterations

$$x_{k+1} = x_k - \frac{\gamma_k}{n} \sum_{i=1}^{n} \mathsf{clip}_\tau(\nabla f_i(x_k)), \tag{4}$$

Table 1: Known and our new results for first order methods with clipping. $^{(a)}$ It is not clear if the analysis is correct: `https://openreview.net/forum?id=hq7vLjZTJPk`. $^{(b)}$ Relies on bounded gradient and gradient similarity assumptions: there exists $G \geq 0$ and $\sigma_g \geq 0$ such that $\|\nabla f_i(x)\|^2 \leq G^2$ and $\|\nabla f_i(x) - \nabla f(x)\|^2 \leq \sigma_g^2$ for all $x \in \mathbb{R}^d$ and all $i \in [n]$. $^{(c)}$ considers local steps for communication efficiency, but these lead to a worse communication efficiency guarantee. $^{(d)}$ $\sigma^2$ is the variance of the Gaussian noise.

| Algorithm | Covers $n > 1$ Case? | Nonconvex Rate | Communication Compression | Comment |
|---|---|---|---|---|
| Clip-GD Zhang et al. (2020a) | ✗ | $\mathcal{O}(1/K)$ | — | $(L_0', L_1')$-smoothness applies to $n = 1$ case only |
| CE-FedAvg Zhang et al. (2022) | ✓ | $\mathcal{O}(1/\sqrt{K})$ | ✗ $^{(c)}$ | strong assumptions $^{(b)}$ slow rate |
| CELGC Liu et al. (2022) | ✓ | $\mathcal{O}(1/\sqrt{K})$ $^{(a)}$ | ✗ | strong assumptions $^{(b)}$ slow rate |
| Clip21-Avg NEW (Alg 1) | ✓ | $\mathcal{O}(\max\{0, 1 - K\})$ (Thm 4.3) | ✗ | Exact solution in $O(1)$ steps Solves average estimation |
| Clip21-GD NEW (Alg 2) | ✓ | $\mathcal{O}(1/K)$ (Thm 5.6) | ✗ | Fast GD-like rate under $L$-smoothness |
| Press-Clip21-GD NEW (Alg 3) | ✓ | $\mathcal{O}(1/K)$ (Thm L.1) | ✓ | Fast GD-like rate under $L$-smoothness |

where $\gamma_k > 0$ is a stepsize, and $\mathsf{clip}_\tau : \mathbb{R}^d \to \mathbb{R}^d$ is the clipping operator with threshold $\tau > 0$, defined via

$$\mathsf{clip}_\tau(x) := \begin{cases} x & \text{if} \quad \|x\| \leq \tau \\ \frac{\tau}{\|x\|}x & \text{if} \quad \|x\| > \tau \end{cases}. \tag{5}$$

## 1.2 Convergence of Clip-GD in the $n = 1$ case

In the $n = 1$ case, Clip-GD was studied by Zhang et al. (2020a), where it was shown that, under our assumptions[1], and for suitable stepsizes,

$$\|\nabla f(x_K)\|^2 \leq \frac{20L(f(x_0) - f_{\inf})}{K}.$$

This is the same rate as that of vanilla GD, i.e., the clipping bias does not cause any convergence issues in the $n = 1$ regime. Zhang et al. (2020a) did not consider the distributed ($n > 1$) regime since their work was motivated by orthogonal considerations to ours: instead of tackling the issue of clipping bias inherent to the distributed regime, as we do, they set out to explain the efficacy of clipping as a tool for convergence stabilization of GD for functions with rapidly growing gradients (this issue exists even in the single node regime). To model functions with rapidly growing gradients, they consider the $(L_0', L_1')$-smoothness assumption, which requires the bound $\|\nabla^2 f(x)\| \leq L_0' + L_1' \|\nabla f(x)\|$ to hold for all $x \in \mathbb{R}^d$. For twice continuously differentiable functions, this assumption specializes to ours by setting $L_0' = L = L_1$ and $L_1' = 0$.

## 1.3 Divergence of Clip-GD in the $n > 1$ case

Surprisingly little is known about the convergence properties of Clip-GD in the $n > 1$ case. The key reason behind this is the poor quality of

$$g(x) := \frac{1}{n} \sum_{i=1}^{n} \mathsf{clip}_\tau(\nabla f_i(x)) \tag{6}$$

as an estimator of $\nabla f(x)$. This issue does not arise in the $n = 1$ case since clipped gradient retains the directional information of the original gradient and all that is lost is just some of the scaling information, which in the light of the results of Zhang et al. (2020a) described above is not problematic. However, the situation is more complicated in the $n > 1$ case as illustrated in the following example.

**Example 1.1.** Let $d = 1$, $n = 2$ with $f_1(x) = \frac{\beta}{2}x^2$ and $f_2(x) = -\frac{\alpha}{2}x^2$, where $\beta > \alpha \geq \tau > 0$ and $x > 0$. Both of these functions have Lipschitz gradients, and $f$ has $(\beta - \alpha)$-Lipschitz gradient.

---

[1]Zhang et al. (2020a) need to additionally assume $f$ to be twice continuously differentiable.

Moreover, $f(x) = \frac{\beta - \alpha}{2} x^2$ is lower bounded below by $f_{\inf} = 0$. So, this setup satisfies our assumptions. Notice that while the gradient is equal to $\nabla f(x) = \frac{1}{2} \nabla f_1(x) + \frac{1}{2} \nabla f_1(x) = \beta x - \alpha x = (\beta - \alpha) x$, the gradient estimator in Equation 6 gives $g(x) = \frac{1}{2} \mathsf{clip}_\tau (\nabla f_1(x)) + \frac{1}{2} \mathsf{clip}_\tau (\nabla f_2(x)) = \min\{1, \frac{\tau}{|\beta x|}\} \beta x + \min\{1, \frac{\tau}{|-\alpha x|}\} (-\alpha x) = \tau - \tau = 0$ whenever $x \geq \frac{\tau}{\alpha}$. This means that Clip-GD will not progress at all if initialized with $x_0 \geq \frac{\tau}{\alpha}$. For example, if we choose $x_0 = \frac{\tau}{\alpha}$, then both the function value $f(x_0) = \frac{\beta - \alpha}{2} \frac{\tau^2}{\alpha^2}$ and the gradient $\nabla f(x_0) = (\beta - \alpha) \frac{\tau}{\alpha}$ can be arbitrarily large (by fixing $\tau$ and $\alpha$, and increasing $\beta$), while the optimal value and gradient of $f$ are both zero.

From this example Clip-GD is a fundamentally flawed method in the $n > 1$ case, unable to converge from many starting points to *any* finite degree of accuracy, however weak accuracy requirements we may have! This is true for any class of functions which includes the above example, and hence it is true for the class of functions we consider in this paper: lower bounded, with Lipschitz gradient.

## 2 SUMMARY OF CONTRIBUTIONS

In the light of the above discussion, there are at least three ways forward: i) consider a different algorithm, ii) consider a different function class, or iii) change both the algorithm and the function class. In this work we explore the first of these three possible approaches: we design a new way of combining clipping and gradient descent – one that does not suffer from any convergence issues.

### 2.1 Clip21-Avg: ERROR FEEDBACK FOR AVERAGE ESTIMATION WITH CLIPPING

As the first step in our process of discovery of fixing the bias caused by clipping in Equation 6 in the gradient estimation, we first study a simplified setting void of any optimization aspect, thus removing one source of dynamics, which greatly simplifies the situation. In particular, we study the problem of estimating the average of *fixed* vectors $a^1, \ldots, a^n \in \mathbb{R}^d$ via repeated use of clipping.

We propose an error feedback mechanism for this task, leading to method Clip21-Avg (see Algorithm 1), and prove that our method finds the *exact* average in

$$K = \mathcal{O}(1/\tau)$$

iterations (see Theorem 4.3). As a corollary, if $\tau$ is sufficiently large, then Clip21-Avg finds the exact average in a single iteration, which is to be expected from any reasonable mechanism. In particular, Clip21-Avg maintains a collection of auxiliary iterates $v_k^1, \ldots, v_k^n$ used to estimate the vectors $a^1, \ldots, a^n$ which we evolve via the rule

$$v_k^i = v_{k-1}^i + \mathsf{clip}_\tau (a^i - v_{k-1}^i), \quad i \in [n]. \tag{7}$$

The average of the vectors is then estimated by $v_k := \frac{1}{n} \sum_{i=1}^n v_k^i \approx \frac{1}{n} \sum_{i=1}^n a^i$.

### 2.2 Clip21-GD: ERROR FEEDBACK FOR GD WITH CLIPPING

Having solved the simpler task of average estimation, we now use similar ideas to design an error feedback mechanism for fixing the bias caused by clipping in an optimization setting for solving Equation 1. In particular, we propose to estimate the average of the gradients $\nabla f_i(x^k), \ldots, \nabla f_n(x^k)$ by the output of a *single* iteration of Clip21-Avg, i.e., by $v_k = \frac{1}{n} \sum_{i=1}^n v_k^i$, where

$$v_k^i = v_{k-1}^i + \mathsf{clip}_\tau (\nabla f_i(x_k) - v_{k-1}^i), \quad i \in [n], \tag{8}$$

and use this estimator in lieu of the true gradient to progress in our optimization task: $x_{k+1} = x_k - \gamma v_k$. This approach leads to the main method of this paper: Clip21-GD (see Algorithm 2). The re-introduction of gradient dynamics into the picture causes considerable issues in that our analysis can not rely on the arguments used to analyze Clip21-Avg. Indeed, while the vectors $a^1, \ldots, a^n$ whose average we are trying to estimate in Clip21-Avg remain static throughout the iterations of Clip21-Avg, in Clip21-GD they change after every step. Analyzing the combined dynamics of two these methods turned out challenging, but ultimately possible, and the result is satisfying. In particular, in Theorem 5.6 we prove that Clip21-GD enjoys the same rate as vanilla GD; that is, our method outputs a (random) point $\hat{x}_K$ such that

$$\mathbf{E}\left[\|\nabla f(\hat{x}_K)\|^2\right] \leq \mathcal{O}(1/K).$$

Our main result can be found in Section 5; see also Table 1.

The closest competing method to ours is that of Zhang et al. (2022), who study the convergence of a clip-enabled variant of the FedAvg algorithm, called CE-FedAvg, under the additional assumptions that there exists $G \geq 0$ and $\sigma_g \geq 0$ such that $\|\nabla f_i(x)\|^2 \leq G^2$ and $\|\nabla f_i(x) - \nabla f(x)\|^2 \leq \sigma_g^2$ for all $x \in \mathbb{R}^d$ and all $i \in [n]$. Given the discussion from the beginning of Section 2, their work can thus be seen as embarking on approach ii) towards taming the divergence issues associated with applying gradient clipping in the $n > 1$ case. We wish to note that these additional assumptions are rather strong. First, it is unclear why gradient clipping is required in the regime when the gradients are already bounded. Second, the function similarity assumption typically does not hold in practice (Khaled et al., 2020; Richtárik et al., 2021). Finally, none of these assumptions hold even for convex quadratics. Even with these additional and arguably strong assumptions, Zhang et al. (2022) in their Theorem 3.1 establish a result of the form

$$\|\nabla f(\hat{x}_K)\|^2 \leq \mathcal{O}(1/\sqrt{K}),$$

which is weaker than the $\mathcal{O}(1/K)$ rate we achieve.

### 2.3 EXTENSION TO ADDING COMMUNICATION COMPRESSION FOR INCREASED COMMUNICATION EFFICIENCY

We extend Clip21-GD to communication-efficient learning applications. We call this method Press-Clip21-GD where each node compresses the clipped vector before it communicates. The description and convergence theorem of Press-Clip21-GD are provided in Appendix L.

### 2.4 EXPERIMENTS

Our experiments on regression and deep learning problems suggest that Clip21-GD substantially outperforms Clip-GD. See Section 6.

## 3 RELATED WORK

**Relation to literature on exploding gradients.** Gradient clipping and normalization were studied in early subgradient optimization literature, e.g., by Shor (1985) and Ermoliev (1988), as techniques for enforcing convergence when minimizing rapidly growing (i.e., non-Lipschitz) functions. In contrast to this and also more recent literature on this topic Pascanu et al. (2013); Goodfellow et al. (2016); Zhang et al. (2020a); Mai & Johansson (2021), we assume $L$-Lipschitzness of the gradient. This is because the issue we are trying to overcome in our work exists even in this more restrictive regime. In other words, we are not employing clipping as a tool for taming the exploding gradients problem, and our work is fully complementary to this literature. Citing Abadi et al. (2016), "gradient clipping of this form is a popular ingredient of SGD for deep networks for non-privacy reasons, though in that setting it usually suffices to clip after averaging." Clipping after averaging does not cause the severe bias and divergence issues we are addressing in our work.

**Relation to literature on heavy-tailed noise.** In contrast to the literature on using clipping to tame stochastic gradient estimators with a heavy-tailed behavior Nazin et al. (2019); Zhang et al. (2020b); Gorbunov et al. (2020a; 2021); Cutkosky & Mehta (2021), we do not consider the heavy-tailed setup in our work. In fact, our key methods and results are fully meaningful in the deterministic gradient regime, which is why we focus on it in much of the paper.

**Relation to literature on Byzantine robustness.** In our work we do *not* consider the Byzantine setup and instead focus on standard distributed optimization with nodes whose outputs can be trusted. However, there is a certain similarity between our Clip21-Avg method and the centered clipping mechanism employed by Karimireddy et al. (2021) to obtain a Byzantine-robust estimator of the gradient. We shall comment on this in Section 4.

**Relation to literature on error feedback.** Error feedback (EF), originally proposed by Seide et al. (2014), is a popular mechanism for stabilizing optimization methods that use compressed

---

**Algorithm 1** Clip21-Avg (Error Feedback for Average Estimation with Clipping)

---

1: **Input:** initial shifts $v^1_{-1}, \ldots, v^n_{-1} \in \mathbb{R}^d$; clipping threshold $\tau > 0$
2: **for** $k = 0, 1, 2, \ldots, K - 1$ **do**
3:    **for** each worker $i = 1, \ldots, n$ in parallel **do**
4:       $v^i_k = v^i_{k-1} + \mathsf{clip}_\tau(a^i - v^i_{k-1})$
5:    **end for**
6:    $v_k = \frac{1}{n} \sum_{i=1}^n v^i_k$                              $\diamond$ estimate of $a := \frac{1}{n} \sum_i a^i$
7: **end for**

---

gradients to reduce communication costs. Variants of EF methods were originally analyzed by Alistarh et al. (2018); Stich et al. (2018); Wu et al. (2018) and later refined by Tang et al. (2019); Karimireddy et al. (2019); Stich & Karimireddy (2020); Qian et al. (2021); Khirirat et al. (2020). The current best results can be found in (Gorbunov et al., 2020b; Qian et al., 2021). However, these methods were analyzed either in the single-node setting, or homogeneous data setting, or otherwise suffer from restrictive assumptions (e.g., bounded gradient-norm and bounded data dissimilarity conditions) and not fully satisfying rates (e.g., $\mathcal{O}(1/K^{2/3})$ in the nonconvex regime). To address these problems, a new error-feedback mechnism called EF21 was proposed by Richtárik et al. (2021), and shown to provide fast $\mathcal{O}(1/K)$ convergence for distributed optimization over smooth, heterogeneous objective functions Richtárik et al. (2021); Fatkhullin et al. (2021); Richtárik et al. (2022), under weak assumptions. Our algorithmic approach behind Clip21-GD is inspired by the error feedback mechanism EF21 of Richtárik et al. (2021) proposed in the context of distributed optimization with contractive communication compression, but needs a different theoretical approach due to a difference between the properties of the clipping and compression operators which necessitates a substantially more refined and involved analysis. We shall comment on this in more detail in Section 5.

## 4   Error Feedback for Average Estimation with Clipping

We shall now describe the properties of our Clip21-Avg method (Algorithm 1) for finding the average of $n$ vectors, $a^1, \ldots, a^n \in \mathbb{R}^d$.

### 4.1   Basic properties of the clipping operator

It is easy to verify that $\mathsf{clip}_\tau$ is the projection operator onto the ball $\mathcal{B}(0, \tau) := \{x : \|x\| \le \tau\}$, and that is satisfies the properties described in the next lemma.

**Lemma 4.1.** *The clipping operator* $\mathsf{clip}_\tau : \mathbb{R}^d \to \mathbb{R}^d$ *has the following properties for all* $\tau > 0$:

  *(i)* $\mathsf{clip}_{\gamma\tau}(x) = \gamma\mathsf{clip}_\tau\left(\frac{x}{\gamma}\right)$ *for all* $x \in \mathbb{R}^d$ *and* $\gamma > 0$,

  *(ii)* $\|\mathsf{clip}_\tau(x) - x\| = 0$ *if* $\|x\| \le \tau$,

  *(iii)* $\|\mathsf{clip}_\tau(x) - x\| = \|x\| - \tau$ *if* $\|x\| \ge \tau$,

  *(iv)* $\|\mathsf{clip}_\tau(x) - x\|^2 = \left(1 - \frac{\tau}{\|x\|}\right)^2 \|x\|^2$ *if* $\|x\| \ge \tau$.

We will use parts (ii)-(iii) of this lemma in the rest of this section. Part (iv) will be useful in Section 5.

### 4.2   Estimating $a^i$

We now analyze Step 3 of Algorithm 1 (i.e., Equation 7). As we shall see, $v^i_k$ converges to $a^i$ exactly in the finite number of steps.

**Lemma 4.2.** *For all iterates* $k \ge 0$ *of Algorithm 1,*

$$\left\|v^i_k - a^i\right\| \le \max\left\{0, \left\|v^i_{-1} - a^i\right\| - (k+1)\tau\right\}, \forall i. \tag{9}$$

*In particular, if* $\left\|v^i_{-1} - a^i\right\| \le \tau$, *then* $v^i_0 = a^i$. *Otherwise, if* $k \ge \left\lceil \frac{1}{\tau}\left\|v^i_{-1} - a^i\right\| - 1\right\rceil$, *then* $v^i_k = a^i$.

---

**Algorithm 2** Clip21-GD (Error Feedback for Distributed Optimization with Clipping)

1: **Input:** initial iterate $x_0 \in \mathbb{R}^d$; learning rate $\gamma > 0$; initial gradient shifts $v_{-1}^1, \ldots, v_{-1}^n \in \mathbb{R}^d$; clipping threshold $\tau > 0$
2: **for** $k = 0, 1, 2, \ldots, K - 1$ **do**
3:     Broadcast $x_k$ to all workers
4:     **for** each worker $i = 1, \ldots, n$ in parallel **do**
5:         Compute $g_k^i = \mathsf{clip}_\tau(\nabla f_i(x_k) - v_{k-1}^i)$
6:         Update $v_k^i = v_{k-1}^i + g_k^i$
7:     **end for**
8:     $v_k = v_{k-1} + \frac{1}{n} \sum_{i=1}^n g_k^i$
9:     $x_{k+1} = x_k - \gamma \frac{1}{n} \sum_{i=1}^n v_k^i$
10: **end for**

---

## 4.3   Estimating $a := \frac{1}{n} \sum_{i=1}^n a^i$

A convergence result for $v_k \to a$ follows by applying Lemma 4.2 for all $i \in [n]$ and using convexity of the norm.

**Theorem 4.3.** *For all iterates $k \geq 0$ of Algorithm 1,*

$$\|v_k - a\| \leq \frac{1}{n} \sum_{i=1}^n \max \left\{ 0, \left\| v_{-1}^i - a^i \right\| - (k+1)\tau \right\}.$$

*In particular, if $\left\| v_{-1}^i - a^i \right\| \leq \tau$ for all $i$, then $v_0 = a$. Otherwise, if $k \geq \max_i \left\lceil \frac{1}{\tau} \left\| v_{-1}^i - a^i \right\| - 1 \right\rceil$, then $v_k = a$.*

## 5   Error Feedback for Distributed Optimization with Clipping

The design of our new method Clip21-GD (Algorithm 2) is inspired by the current state-of-the-art error feedback mechanism called EF21 developed by Richtárik et al. (2021) (see (Fatkhullin et al., 2021; Richtárik et al., 2022) for extensions) the goal of which is to progressively remove the error introduced by a *contractive compression operator* applied to the gradients[2]. A contractive operator is a possibly randomized mapping $\mathcal{C} : \mathbb{R}^d \to \mathbb{R}^d$ satisfying the property

$$\mathbf{E} \left[ \|\mathcal{C}(x) - x\|^2 \right] \leq (1 - \alpha) \|x\|^2, \quad \forall x \in \mathbb{R}^d \tag{10}$$

for some $0 < \alpha \leq 1$. However, the results of Richtárik et al. (2021) do not apply to our setup since the clipping operator is not contractive in the sense of Equation 10. Our idea is to instead rely on the related identity

$$\|\mathsf{clip}_\tau(x) - x\|^2 = \begin{cases} 0 & \text{if } \|x\| \leq \tau \\ \left(1 - \frac{\tau}{\|x\|}\right)^2 \|x\|^2 & \text{if } \|x\| \geq \tau \end{cases}$$

established in Lemma 4.1. Using this identity in lieu of Equation 10 is more complicated since the contraction factor can be arbitrarily if $\|x\|$ is large. We needed to develop a new analysis technique to handle this situation.

In the rest of this section we describe the strong theoretical properties of Clip21-GD (Algorithm 2).

### 5.1   Single-node regime ($n = 1$)

We begin by studying Clip21-GD in the single-node case. For simplicity, in the subsequent text and statements, we drop the superscript $i$ from all iterates.

In the light of the discussion from Section 1.2, in this case our error feedback mechanism for clipping is *not* needed; that is, Clip-GD suffices, and there is no need for Clip21-GD. However, one would hope that our approach offers comparable guarantees in this case to those obtained by Zhang et al. (2020a) in the $L$-smooth regime; i.e., we expect to obtain a $\mathcal{O}(1/K)$ rate. The key purpose of this section is to

---

[2]This is why use the number 21 in the name Clip21.

see that this is indeed the case. However, we believe that Clip21-GD *is* needed even in the $n = 1$ case if one wants to obtain results in the more general constrained or proximal regime[3], and hence the results of this section can serve as a basis for further exploration and extensions.

Our first result establishes a descent lemma for a certain Lyapunov function. This is a substantial departure from existing analyses of clipping methods which do not make use of the control variate sequence $\{v_k\}$.

**Lemma 5.1** (Descent lemma). *Consider the problem of minimizing $f : \mathbb{R}^d \to \mathbb{R}$, assuming it is $L$-smooth and lower bounded by $f_{\text{inf}} \in \mathbb{R}$. Let $v_{-1} = 0 \in \mathbb{R}^d$, $\eta := \min\left\{1, \frac{\tau}{\|\nabla f(x_0)\|}\right\}$, $F_0 := f(x_0) - f_{\text{inf}}$, and $G_0 := |\|\nabla f(x_0)\| - \tau|$. Then, single-node Clip21-GD (described in Algorithm 2 with $n = 1$) with stepsize*

$$\gamma \leq \frac{1}{L} \min\left\{1 - \frac{1}{\sqrt{2}}, \frac{1}{1+\sqrt{1+2\beta_1}}, \frac{\tau^2}{4L\left[\sqrt{F_0}+\sqrt{\beta_2}\right]^2}\right\}, \tag{11}$$

*where $\beta_1 := \frac{(1-\eta)^2(1+2/\eta)}{1-(1-\eta)(1-\eta/2)}$ and $\beta_2 := F_0 + \frac{\tau G_0}{\sqrt{2\eta}L}$, satisfies*

$$\phi_{k+1} \leq \phi_k - \frac{\gamma}{2}\|\nabla f(x_k)\|^2, \tag{12}$$

*where $\phi_k := f(x_k) - f_{\text{inf}} + A\|\nabla f(x_k) - v_k\|^2$ and $A := \frac{\gamma}{2[1-(1-\eta)(1-\eta/2)]}$.*

This lemma states that the Lyapunov function $\phi_k$ in Equation 12 decreases in each iteration by an amount proportional to the squared norm of the gradient, regardless of whether the clipping operator is active or not. We next prove the state of the clipping operator when the algorithm is run. Our next result states that if the clipping operator is "active" at the start (i.e., $\|\nabla f(x_0)\| > \tau$), it will become "inactive" (i.e., it will act as the identity mapping) after at most $\mathcal{O}(\|\nabla f(x_0)\|/\tau)$ iterations, and then stay inactive from then on.

**Proposition 5.2** (Finite-time to no clipping). *Let the conditions of Lemma 5.1 hold.*

(i) *If $x_0$ satisfies $\|\nabla f(x_0)\| \leq \tau$, then $\|\nabla f(x_k) - v_{k-1}\| \leq \tau$ for all $k \geq 0$. That is, the clipping operator is inactive for all iterations.*

(ii) *If $x_0$ satisfies $\|\nabla f(x_0)\| > \tau$, then $\|\nabla f(x_k) - v_{k-1}\| \leq \tau$ for $k \geq k^\star := \frac{2}{\tau}(\|\nabla f(x_0)\| - \tau) + 1$. That is, the clipping operator becomes inactive after at most $k^\star = \mathcal{O}(1/\tau)$ iterations.*

Note that when clipping becomes inactive, we have $v_k = v_{k-1} + \text{clip}_\tau(\nabla f(x_k) - v_{k-1}) = \nabla f(x_k)$, which means that Clip21-GD turns into GD after at most $k^\star$ iterations. Finally, Lemma 5.1 and Proposition 5.2 lead to our main convergence result.

**Theorem 5.3** (Convergence result). *Consider single-node Clip21-GD (described in Algorithm 2 with $n = 1$). Let the conditions of Lemma 5.1 hold and let $\hat{x}_K$ be a point selected from the set $\{x_0, x_1, \ldots, x_{K-1}\}$ for $K \geq 1$ uniformly at random. Then*

$$\mathbf{E}\left[\|\nabla f(\hat{x}_K)\|^2\right] \leq \frac{2}{\gamma}\frac{\phi_0}{K}.$$

*If $\gamma$ is chosen to be the right-hand side of Equation 11, then*

$$\mathbf{E}\left[\|\nabla f(\hat{x}_K)\|^2\right] = \mathcal{O}\left(\left[\frac{C(\|\nabla f(x_0)\|+\tau)}{\tau} + \frac{L^2(F_0)^2}{\tau^2}\right]\frac{1}{K}\right),$$

*where $C := \max\left\{LF_0, \|\nabla f(x_0)\|^2\right\}$.*

Theorem 5.3 states that in the $L$-smooth non-convex regime, single-node Clip21-GD enjoys the $\mathcal{O}(1/K)$ rate. Up to constant factors, this is the same rate as that of GD.

---

[3]We believe such results can be obtained by using the techniques developed by Fatkhullin et al. (2021).

## 5.2 MULTI-NODE REGIME ($n > 1$)

Next, we turn our attention to multi-node Clip21-GD as described in Algorithm 2. Note that this method becomes EF21 when we replace $\text{clip}_\tau(\cdot)$ with a contractive compressor, and becomes GD when we let $\tau \to +\infty$.

Our results for multi-node Clip21-GD have the same meaning as those presented in the single-node case, and hence a brief commentary comparing these results to the $n = 1$ case should suffice.

**Lemma 5.4** (Descent lemma). *Consider multi-node Clip21-GD (described in Algorithm 2 for general $n$) for solving Equation 1. Suppose that each $f_i(x)$ is $L_i$-smooth and that $f$ is $L$-smooth and lower bounded by $f_{\inf} \in \mathbb{R}$. Let $v^i_{-1} = 0$ for all $i$, $\eta := \min\left\{1, \frac{\tau}{\max_i \|\nabla f_i(x_0)\|}\right\}$, $F_0 := f(x_0) - f_{\inf}$, and $G_0^2 := \frac{1}{n}\sum_{i=1}^n (\|\nabla f_i(x_0)\| - \tau)^2$, with stepsize*

$$\gamma \le \min\left\{ \frac{\eta/L_{\max}}{8}, \frac{(1-1/\sqrt{2})/L}{1+\sqrt{1+2\beta_1}}, \frac{\tau^2/L_{\max}^2}{16\left[\sqrt{F_0}+\sqrt{\beta_2}\right]^2} \right\}, \tag{13}$$

*where $\beta_1 := \frac{(1-\eta)^2(1+2/\eta)}{1-(1-\eta)(1-\eta/2)}\left(L_{\max}/L\right)^2$ and $\beta_2 := F_0 + \frac{\tau G_0}{\sqrt{2\eta}L_{\max}}$. Then*

$$\phi_{k+1} \le \phi_k - \frac{\gamma}{2}\|\nabla f(x_k)\|^2, \tag{14}$$

*where $\phi_k := f(x_k) - f_{\inf} + \frac{A}{n}\sum_{i=1}^n \left\|\nabla f_i(x_k) - v^i_k\right\|^2$ and $A := \frac{\gamma}{2[1-(1-\eta)(1-\eta/2)]}$.*

**Proposition 5.5** (Finite-time to no clipping). *Let the conditions of Lemma 5.4 hold.*

(i) *If $x_0$ satisfies $\|\nabla f_i(x_0)\| \le \tau$, then $\|\nabla f_i(x_k) - v^i_{k-1}\| \le \tau$ for all $k \ge 0$.*

(ii) *If $x_0$ satisfies $\|\nabla f_i(x_0)\| > \tau$, then $\|\nabla f_i(x_k) - v^i_{k-1}\| \le \tau$ for $k \ge k^\star := \frac{2}{\tau}\left(\|\nabla f_i(x_0)\| - \tau\right) + 1$.*

**Theorem 5.6** (Convergence result). *Let the conditions of Lemma 5.4 hold, and $\hat{x}_K$ be a point selected from the set $\{x_0, x_1, \ldots, x_K\}$ uniformly at random for $K \ge 1$. Then, multi-node Clip21-GD (described in Algorithm 2) satisfies*

$$\mathbf{E}\left[\|\nabla f(\hat{x}_K)\|^2\right] \le \frac{2}{\gamma}\frac{\phi_0}{K}.$$

*If $\gamma$ is chosen to be the right-hand side of Equation 13, then*

$$\mathbf{E}\left[\|\nabla f(\hat{x}_K)\|^2\right] = \mathcal{O}\left(\left[\left(1+\frac{C_1}{\tau}\right)C_2 + \frac{L_{\max}^2(F_0)^2}{\tau^2}\right]\frac{1}{K}\right),$$

*where $C_1 := \max_i \|\nabla f_i(x_0)\|$, $C_2 := \max\{L_{\max}F_0, C_1^2\}$.*

Theorem 5.6 says that Clip21-GD enjoys an $\mathcal{O}(1/K)$ rate, which is faster than the previous state-of-the-art rate $\mathcal{O}(1/\sqrt{K})$ rate obtained by Zhang et al. (2022); Liu et al. (2022) on nonconvex problems. Also, we do not require bounded gradient and function similarity assumptions as they do (see Table 1). Furthermore, Proposition 5.5 says that if the clipping operator at each node is "active" at the start (i.e., $\|\nabla f_i(x_0)\| > \tau$), it will become "inactive" in at most $k^\star$ steps (i.e., $\|\nabla f_i(x_k) - v^i_{k-1}\| \le \tau$, and Clip21-GD will effectively become GD. Moreover, when specializing our multi-node theory for Clip21-GD to the $n = 1$ case, and compare this to the theory from Section 5.1, we pay the price of a smaller maximum stepsize by a factor of $\approx 4$. We comment more on this in Appendix C.

To reduce the communication costs, we further modify Clip21-GD by replacing $\text{clip}_\tau(\nabla f_i(x_k) - v^i_{k-1})$ with $\mathcal{C}(\text{clip}_\tau(\nabla f_i(x_k) - v^i_{k-1}))$, where $\mathcal{C} : \mathbb{R}^d \to \mathbb{R}^d$ is a contractive compressor. This method, which we call Press-Clip21-GD, is shown to enjoy the $\mathcal{O}(1/K)$ rate as well. The method and theory are relegated to Appendix L.

## 6 EXPERIMENTS

To demonstrate strong performance of Clip21-GD over traditional clipped gradient methods, we evaluate all the methods on the regularized logistic regression problem

$$\min_{x \in \mathbb{R}^d}\left[f(x) := \frac{1}{n}\sum_{i=1}^n f_i(x) + \lambda r(x)\right], \quad \text{where} \quad f_i(x) := \frac{1}{m}\sum_{j=1}^m \log(1 + e^{-b_{ij}a_{ij}^\top x})$$

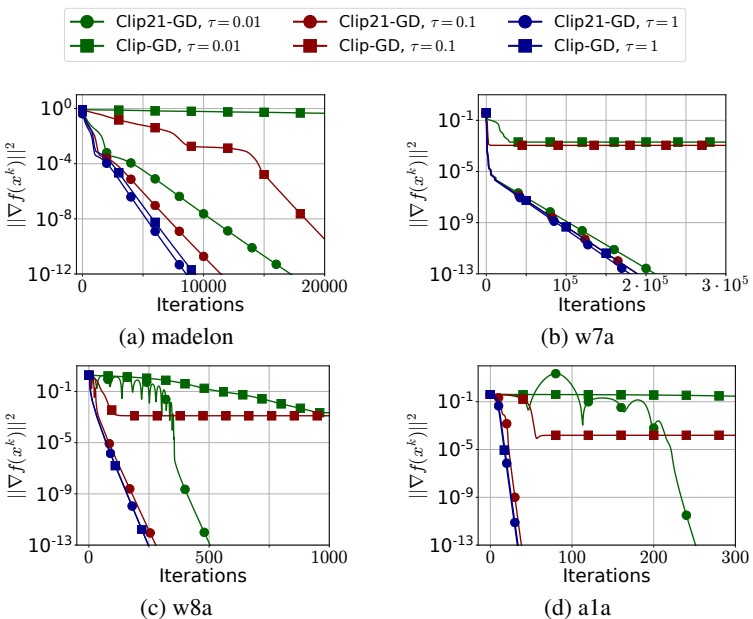

Figure 1: Comparison of Clip21-GD vs Clip-GD with clipping threshold $\tau \in \{0.01, 0.1, 1\}$ on logistic regression with $\ell_2$-regularizer (**first line**) and a nonconvex regularizer (**second line**).

and $a_{ij} \in \mathbb{R}^d$ is the $j^{\text{th}}$ training sample associated with class label $b_{ij} \in \{-1, 1\}$, which is privately known by node $i$. Additional experiments on nonconvex linear regression and deep neural network training are deferred to Section B. We use datasets from the LibSVM library (Chang & Lin, 2011), and two types of regularization: (1) $r(x) = \|x\|^2/2$, which is the $\ell_2$-regularizer, and (2) $r(x) = \sum_{j=1}^{d} x_j^2/(1 + x_j^2)$, which is a nonconvex regularizer. Before training, we preprocess each dataset as follows: we ($i$) sort the training samples according to the labels; ($ii$) split the dataset into $n$ equal parts among the nodes; and ($iii$) normalize each sample of each part by using StandardScaler from the scikit-learn library (Pedregosa et al., 2011). By this preprocessing, the problem becomes more heterogeneous while the Lipschitz constants of functions $\nabla f_i$ are closer to each other. We reported the best convergence performance of the baseline Clip-GD and Clip21-GD by choosing stepsizes $\gamma \in \{1/4L, 1/2L, \ldots, 8/L\}$. We also set $n = 10$; $\lambda = 10^{-1}$ and $\lambda = 10^{-4}$ for nonconvex and $\ell_2$ regularizers respectively.

Figure 1 shows that Clip21-GD outperforms Clip-GD in both the convergence speed and solution accuracy. This happens because when the clipping operator is turned on at the beginning, it will be turned off in Clip-GD with required iteration counts $k$ much larger than in Clip21-GD for small values of $\tau$. For $\tau \in \{0.01, 0.1\}$, Clip-GD converges towards the neighborhood while Clip21-GD always converges towards the stationary point. At iteration $k = 10^4$ and for $\tau = 0.01$, Clip21-GD achieves roughly 6 times more accurate solution than Clip-GD for the madelon and w7a datasets.

## 7 CONCLUSIONS, LIMITATIONS AND EXTENSIONS

We proposed Clip21-GD – an error feedback mechanism for dealing with the biased caused by gradient clipping. We proved that Clip21-GD enjoys the $\mathcal{O}(1/K)$ convergence rate for nonconvex optimization in single-node and multi-node settings. Our numerical experiments indicate that Clip21-GD attains faster convergence speed and higher solution accuracy than Clip-GD. We plan to extend our theory for Clip21-GD to stochastic optimization as it works well in our experiments on training deep neural network models.

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

## A  RELATION TO LITERATURE ON BYZANTINE ROBUSTNESS

Algorithm 1 is similar to the *centered clipping* subroutine used by (Karimireddy et al., 2021) to obtain a Byzantine-robust estimator of the gradient. In their setting, the clients are partitioned into two groups, regular (majority) and Byzantine (minority), and the goal is to minimize the average of the functions owned by the regular clients without a-priori knowing which clients are regular. The Byzantine clients are allowed to report any vectors in an adversarial fashion in an attempt to induce bias into gradient estimation. In this application, it is assumed that the regular workers share the same function as this ensures that there is enough "signal" for the optimization method to iteratively find out which clients are regular. In contrast, we allow all functions $f_i$ to be arbitrarily heterogeneous. While (Karimireddy et al., 2021) use a single shared center/shift for all clients $i \in [n]$ the purpose of which is to "learn" who the regular (i.e., non-Byzantine) clients are via tracking the (homogeneous) gradient of these regular clients, we use $n$ different centers/shifts $(v_k^1, \ldots, v_k^n)$ designed to track and ultimately find all the original vectors $a^1, \ldots, a^n$, respectively, which can be arbitrarily different. Due to the different goal they have, their analysis is completely different to ours.

## B  ADDITIONAL EXPERIMENTS

We now include additional results on nonconvex linear regression and deep learning tasks.

### B.1  NONCONVEX LINEAR REGRESSION

We run all clipped methods to solve the linear regression problem with the nonconvex regularization on the form:

$$\min_{x \in \mathbb{R}^d} \left[ f(x) = \frac{1}{n} \sum_{i=1}^{n} f_i(x) \right],$$

where each local loss function is

$$f_i(x) = \frac{1}{m} \sum_{j=1}^{m} (a_{ij}^\top x - b_{ij})^2 + \lambda \sum_{j=1}^{d} \frac{x_j^2}{1 + x_j^2}.$$

Here, $\lambda > 0$ is the nonconvex regularization parameter, and $(a_{i1}, b_{i1}), \ldots, (a_{im}, b_{im})$ where $a_{ij} \in \mathbb{R}^d$ and $b_{ij} \in \{-1, 1\}$ are $m$ training data samples available for node $i$. We use datasets from LibSVM library (Chang & Lin, 2011), and perform the preprocessing steps described in Section 6. These preprocessing steps allow some workers to have data samples with only one class label, which make the problem more heterogeneous. Also, by the normalization step of this preprocessing Lipschitz constants of local loss functions $f_i(x)$ become close to each other. We also used the same set of parameters (i.e. $n, \lambda, \tau, \sigma$) for the linear regression problem as the logistic regression problem.

From Figure 2 (a)-(b), the convergence speed of Clip21-GD is faster than or the same as Clip-GD. For instance, Clip21-GD converges faster than Clip-GD when $\tau = 0.1$ while both methods have the same convergence performance when $\tau = 1$. This happens because when the clipping operator is turned on at the beginning, it will be turned off in Clip-GD with required iteration counts $k$ much larger than in Clip21-GD for small values of $\tau$ (e.g. for $\tau = 0.1$, Clip-GD does not converge at all).

### B.2  DEEP LEARNING EXPERIMENTS

We now showcase that Clip21 also outperforms Clip for training the VGG11 model (Simonyan & Zisserman, 2014) for multiclass classification problem on the CIFAR10 train dataset with $50000$ samples and 10 classes (5000 samples for each class) (Krizhevsky et al., 2009). We modify Clip21-GD and Clip-GD by replacing the full local gradient $\nabla f_i(x_k)$ with its mini-batch stochastic estimator. We refer these methods as Clip21-SGD and Clip-SGD.

This data set is split into 10 classes among 10 workers according to the following rules:

1. $2500$ samples of the $i^{th}$ class are given to the $i^{th}$ client (which is in total $25000$ samples), and

2. the rest of dataset is shuffled and partitioned randomly between workers.

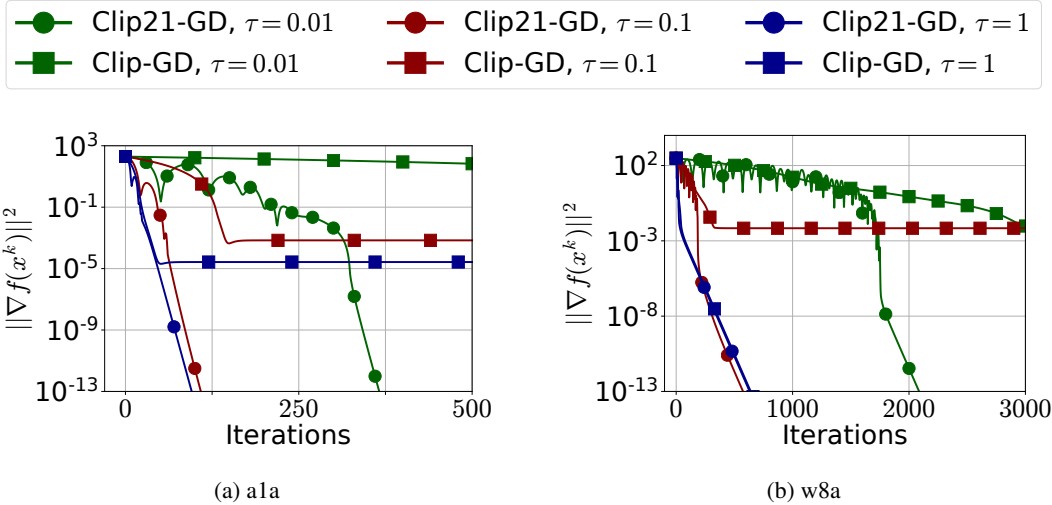

(a) a1a

(b) w8a

Figure 2: (a), (b) The comparison of Clip-EF21 and Clip-GD varying clipping parameter $\tau$ on Linear Regression with nonconvex regularization.

These preprocessing rules allow the $i^{\text{th}}$ worker to have most samples with the $i^{\text{th}}$ class, which makes the problem more heterogeneous. For each worker, its local datasets are shuffled only once at the beginning, and its stochastic gradient is computed for each iteration from randomly selected samples with the batch size 32.

We reported the best performance of Clip21-SGD and Clip-SGD in train loss and test accuracy from fine-tuning stepsizes (see Figure 3). We observe that Clip21-SGD outperforms Clip-SGD in both metrics for any values of $\tau$; see Figure 3. In particular, one can notice that for small value of clipping parameter ($\tau = 10^{-4}$) the difference in train loss and test accuracy given by Clip21-SGD and Clip-SGD is significant, while for relatively large values ($\tau = 10^{-2}$) the performance of algorithms becomes similar. Besides, Clip21-SGD attains more than 3 (in log scale) times lower train loss than Clip-SGD at epoch 50 for $\tau \in \{10^{-4}, 10^{-3}, 10^{-2}\}$. These encouraging experiments motivate us to investigate theoretical convergence guarantees for Clip21-SGD as our future directions.

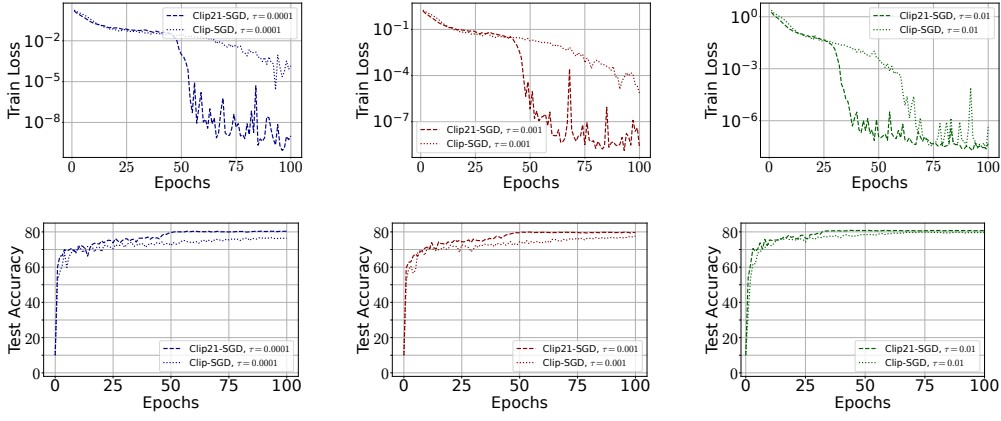

Figure 3: The performance of Clip21-SGD and Clip-SGD with fine-tuned stepsizes to train the VGG11 model on the CIFAR10 dataset.

## C  LOSS OF A CONSTANT FACTOR WHEN GENERALIZING TO ARBITRARY $n$

When specializing our multi-node theory for Clip21-GD to the $n = 1$ case, and compare this to the theory from Section 5.1, we pay the price of a smaller maximum stepsize, which affects constants in the convergence rate.

We illustrate this by letting $\phi_0$ to be large, $L = L_{\max}$ and $n = 1$ in Lemma 5.4, Proposition 5.5 and Theorem 5.6. Suppose that $\frac{\|\nabla f(x_0)\|}{\tau}$ is close to $1$. Then, Lemma 5.4, Proposition 5.5 and Theorem 5.6 with $n = 1$ recover the results from Section 5.1 under the stepsize

$$\gamma \leq \frac{1}{L} \cdot \min \left( \frac{1}{8}, \frac{1 - 1/\sqrt{2}}{1 + \sqrt{1 + 2\beta_1}}, \frac{\tau^2}{16L \left[ \sqrt{F_0} + \sqrt{\beta_2} \right]^2} \right),$$

where $\eta, \beta_1, \beta_2, F_0, G_0$ are defined in Theorem 5.3.

This step-size is $4$ times smaller than that allowed by Theorem 5.3 dedicated to the single-node setting. The technical difficulty preventing us from generalizing the single-node theory to the multi-node version in a tighter manner is related to upper bounding the quantity $\|v_k\|$ in the analysis. In the multi-node setting, we have

$$\|v_k\| = \left\| \nabla f(x_k) + \frac{1}{n} \sum_{i=1}^{n} e_k^i \right\| \leq \|\nabla f(x_k)\| + \frac{1}{n} \sum_{i=1}^{n} \|e_k^i\|,$$

where

$$e_k^i = \mathsf{clip}_\tau(\nabla f_i(x_k) - v_{k-1}^i) - (\nabla f_i(x_k) - v_{k-1}^i).$$

This upper bound on $\|v_k\|$ is looser than that we could use in the single-node setting, which instead reads:

$$\|v_k\| \leq (1 - \eta_k) \|v_{k-1}\| + \eta_k \|\nabla f(x_k)\|,$$

where

$$\eta_k = \min \left\{ 1, \frac{1}{\|\nabla f(x_k) - v_{k-1}\|} \right\}.$$

# D  BASIC INEQUALITIES AND USEFUL LEMMAS

## D.1  BASIC INEQUALITIES

Triangle inequality:

$$\|x + y\| \leq \|x\| + \|y\|, \quad \forall x, y \in \mathbb{R}^d. \tag{15}$$

Subadditivity of the square root:

$$\sqrt{a + b} \leq \sqrt{a} + \sqrt{b}, \quad \forall a, b \geq 0. \tag{16}$$

Young's inequality:

$$\|x + y\|^2 \leq (1 + \theta) \|x\|^2 + (1 + \theta^{-1}) \|y\|^2, \quad \forall x, y \in \mathbb{R}^d, \ \theta > 0. \tag{17}$$

## D.2  LEMMAS

In this section, we introduce several lemmas that are instrumental to our analysis.

**Lemma D.1.** *Let $f : \mathbb{R}^d \to \mathbb{R}$ be a function with $L$-Lipschitz gradient, with $f_{\inf} \in \mathbb{R}$ being a lower bound on $f$. Then*

$$\frac{1}{2L} \|\nabla f(x)\|^2 \leq f(x) - f_{\inf}, \quad \forall x \in \mathbb{R}^d. \tag{18}$$

**Lemma D.2.** *Let $f : \mathbb{R}^d \to \mathbb{R}$ be a function with $L$-Lipschitz gradient. Then*

$$f(x) \leq f(y) + \langle \nabla f(y), y - x \rangle + \frac{L}{2} \|x - y\|^2, \quad \forall x \in \mathbb{R}^d. \tag{19}$$

Lemma D.3 gives the bound on the function value after one step of a method of the type: $x_{k+1} := x_k - \gamma v_k$. We use this lemma to derive the descent inequality for Clip21-GD and its variants.

**Lemma D.3** (Lemma 2 from (Li et al., 2021)). *Let $f : \mathbb{R}^d \to \mathbb{R}$ be a function with $L$-Lipschitz gradient and let $x_{k+1} := x_k - \gamma v_k$, where $\gamma > 0$ and $v_k \in \mathbb{R}^d$ is any vector. Then*

$$f(x_{k+1}) \leq f(x_k) - \frac{\gamma}{2} \|\nabla f(x_k)\|^2 - \left( \frac{1}{2\gamma} - \frac{L}{2} \right) \|x_{k+1} - x_k\|^2 + \frac{\gamma}{2} \|\nabla f(x_k) - v_k\|^2. \tag{20}$$

Finally, we present Lemma D.4 for deriving the easy-to-write condition on the step-size that ensures the convergence of Clip21-GD and its variants.

**Lemma D.4.** *If the stepsize is chosen to satisfy*

$$0 < \gamma \leq \frac{2}{L(\beta_1 + \sqrt{\beta_1^2 + 4\beta_2})}$$

*for some $\beta_1, \beta_2 > 0$, then*

$$\frac{1}{\gamma} - \beta_1 L - \gamma \beta_2 L^2 \geq 0.$$

*Proof.* Let $\gamma = {}^1\!/(\theta L)$ where $\theta > 0$. Then, $\gamma > 0$. Also, ${}^1\!/\gamma - \beta_1 L - \gamma \beta_2 L^2 \geq 0$ can be expressed equivalently as $\theta^2 - \beta_1 \theta - \beta_2 \geq 0$. This condition holds if $\theta \geq {}^{(\beta_1 + \sqrt{\beta_1^2 + 4\beta_2})}\!/2$ or equivalently $0 < \gamma \leq {}^2\!/[L(\beta_1 + \sqrt{\beta_1^2 + 4\beta_2})]$. □

# E   PROOF OF LEMMAS IN SECTION 4

## E.1   PROOF OF LEMMA 4.2

Subtracting $a^i$ from both sides of Step 3 of Algorithm 1, and applying norms, we get

$$
\begin{aligned}
\left\| a^i - v_k^i \right\| &= \left\| v_{k-1}^i + \mathsf{clip}_\tau(a^i - v_{k-1}^i) - a^i \right\| \\
&= \left\| \mathsf{clip}_\tau(a^i - v_{k-1}^i) - (a^i - v_{k-1}^i) \right\|.
\end{aligned}
$$

In view of Lemma 4.1, parts (ii) and (iii), the last expression is 0 if $\left\| a^i - v_{k-1}^i \right\| \leq \tau$, and is equal to $\left\| a^i - v_{k-1}^i \right\| - \tau$ otherwise. The rest follows.

## E.2   PROOF OF THEOREM 4.3

By convexity of the norm, the quantity $\|v_k - a\| = \left\| \frac{1}{n} \sum_{i=1}^n v_k^i - \frac{1}{n} \sum_{i=1}^n a^i \right\|$ can be upper bounded by $\frac{1}{n} \sum_{i=1}^n \left\| v_k^i - a^i \right\|$. It remains to apply Lemma 4.2.

# F    CLIP21-GD IN THE $n = 1$ REGIME

We will now analyze the single-node version of Clip21-GD (Algorithm 2 for $n = 1$), i.e.,

$$x_{k+1} = x_k - \gamma v_k, \tag{21}$$

where

$$v_k = v_{k-1} + \mathsf{clip}_\tau(\nabla f(x_k) - v_{k-1}). \tag{22}$$

Note that this can be rewritten into the simpler form

$$v_k = (1 - \eta_k)v_{k-1} + \eta_k \nabla f(x_k), \tag{23}$$

where

$$\eta_k := \min\left\{1, \frac{\tau}{\|\nabla f(x_k) - v_{k-1}\|}\right\}, \tag{24}$$

and the ratio $\frac{\tau}{\|\nabla f(x_k) - v_{k-1}\|}$ is interpreted as $+\infty$ if $\nabla f(x_k) = v_{k-1}$, which means that $\eta_k = 1$ in that case. Note that this means that

$$\|\nabla f(x_k) - v_k\|^2 \overset{Equation\ 23}{=} (1 - \eta_k)^2 \|\nabla f(x_k) - v_{k-1}\|^2. \tag{25}$$

Recall that the Lyapunov function was defined as

$$\phi_k := f(x_k) - f_{\inf} + \frac{\gamma A_\eta}{2}\|\nabla f(x_k) - v_k\|^2, \tag{26}$$

where $A_\eta := \frac{1}{1 - (1-\eta)\left(1 - \frac{\eta}{2}\right)}$ and $\eta := \min\left\{1, \frac{\tau}{\|\nabla f(x_0)\|}\right\}$. Also let $F_0 := f(x_0) - f_{\inf}$ and $G_0 := \max\{0, \|\nabla f(x_0)\| - \tau\}$.

## F.1    CLAIMS

We first establish several simple but helpful results.

**Claim F.1.** *Assume that $v_{-1} = 0$. Then*

$$v_0 = \mathsf{clip}_\tau(\nabla f(x_0)), \tag{27}$$

$$\|v_0\| = \min\{\tau, \|\nabla f(x_0)\|\}, \tag{28}$$

*and*

$$\|\nabla f(x_0) - v_0\| = \max\{0, \|\nabla f(x_0)\| - \tau\}. \tag{29}$$

*If we additionally assume that $\gamma \leq \frac{2}{L}$, then*

$$\|v_0\| \leq \sqrt{\frac{4}{\gamma}\phi_0}. \tag{30}$$

*Proof.* Equation Equation 27 follows from

$$v_0 \overset{Equation\ 22}{=} v_{-1} + \mathsf{clip}_\tau(\nabla f(x_0) - v_{-1}) = \mathsf{clip}_\tau(\nabla f(x_0)).$$

Equation Equation 28 follows from Equation 27. Equation Equation 29 follows from Equation 27 and Lemma 4.1:

$$\|\nabla f(x_0) - v_0\| \overset{Equation\ 27}{=} \|\nabla f(x_0) - \mathsf{clip}_\tau(\nabla f(x_0))\| \overset{(Lemma\ 4.1(ii)\text{-}(iii))}{=} \max\{0, \|\nabla f(x_0)\| - \tau\}.$$

Finally, Equation 30 follows from

$$
\begin{aligned}
\|v_0\| &\overset{Equation\ 28}{=} \min\{\tau, \|\nabla f(x_0)\|\} \\
&\leq \|\nabla f(x_0)\| \\
&\overset{Equation\ 18}{\leq} \sqrt{2L\left(f(x_0) - f_{\inf}\right)} \\
&\overset{Equation\ 26}{\leq} \sqrt{2L\phi_0} \\
&\leq \sqrt{\frac{4}{\gamma}\phi_0},
\end{aligned}
$$

where the last inequality follows from the assumption $\gamma \leq \frac{2}{L}$. $\qquad\square$

**Claim F.2.** *Fix $k \geq 0$. Then*

$$\|\nabla f(x_{k+1}) - v_k\| \leq \max\{0, \|\nabla f(x_k) - v_{k-1}\| - \tau\} + L\gamma \|v_k\|. \tag{31}$$

*Proof.* Indeed,

$$
\begin{aligned}
\|\nabla f(x_{k+1}) - v_k\| &= &&\|\nabla f(x_k) - v_k + \nabla f(x_{k+1}) - \nabla f(x_k)\| \\
&\overset{Equation\ 15}{\leq} &&\|\nabla f(x_k) - v_k\| + \|\nabla f(x_{k+1}) - \nabla f(x_k)\| \\
&\overset{Equation\ 22}{=} &&\|(\nabla f(x_k) - v_{k-1}) - \mathsf{clip}_\tau(\nabla f(x_k) - v_{k-1})\| \\
& &&+ \|\nabla f(x_{k+1}) - \nabla f(x_k)\| \\
&\overset{(Lemma\ 4.1(ii)\text{-}(iii))}{=} &&\max\{0, \|\nabla f(x_k) - v_{k-1}\| - \tau\} + \|\nabla f(x_{k+1}) - \nabla f(x_k)\| \\
&\overset{Equation\ 3}{\leq} &&\max\{0, \|\nabla f(x_k) - v_{k-1}\| - \tau\} + L\|x_{k+1} - x_k\| \\
&\overset{Equation\ 21}{=} &&\max\{0, \|\nabla f(x_k) - v_{k-1}\| - \tau\} + L\gamma \|v_k\|.
\end{aligned}
$$

$\square$

**Claim F.3.** *Fix $k \geq 1$. If $\frac{\gamma}{2}\|\nabla f(x_{k-1})\|^2 \leq \phi_0$, $\frac{\gamma}{4}\|v_{k-1}\|^2 \leq \phi_0$ and $\gamma \leq \frac{1-\frac{1}{\sqrt{2}}}{L}$, then*

$$\|v_k\| \leq \sqrt{\frac{4}{\gamma}\phi_0}. \tag{32}$$

*Proof.*

$$
\begin{aligned}
\|v_k\| &\overset{Equation\ 23+Equation\ 15}{\leq} &&(1-\eta_k)\|v_{k-1}\| + \eta_k\|\nabla f(x_k)\| \\
&\overset{Equation\ 15}{\leq} &&(1-\eta_k)\|v_{k-1}\| + \eta_k\|\nabla f(x_k) - \nabla f(x_{k-1})\| + \eta_k\|\nabla f(x_{k-1})\| \\
&\overset{Equation\ 3}{\leq} &&(1-\eta_k)\|v_{k-1}\| + \eta_k L\|x_k - x_{k-1}\| + \eta_k\|\nabla f(x_{k-1})\| \\
&\overset{Equation\ 21}{=} &&(1-\eta_k)\|v_{k-1}\| + \eta_k L\gamma\|v_{k-1}\| + \eta_k\|\nabla f(x_{k-1})\| \\
&= &&(1-\eta_k + \eta_k L\gamma)\|v_{k-1}\| + \eta_k\|\nabla f(x_{k-1})\| \\
&\overset{(*)}{\leq} &&(1-\eta_k + \eta_k L\gamma)\sqrt{\frac{4}{\gamma}\phi_0} + \eta_k\sqrt{\frac{2}{\gamma}\phi_0} \\
&= &&\left(1-\eta_k + \eta_k L\gamma + \frac{\eta_k}{\sqrt{2}}\right)\sqrt{\frac{4}{\gamma}\phi_0} \\
&\overset{(**)}{\leq} &&\sqrt{\frac{4}{\gamma}\phi_0},
\end{aligned} \tag{33}
$$

where in (*) we have used the first two assumptions, and in (**) we have used the bound on $\gamma$. $\square$

**Claim F.4.** *Assume that $v_{-1} = 0$. If*

$$0 \leq \gamma \leq \frac{\tau^2}{4L^2\left[\sqrt{F_0} + \sqrt{F_0 + \frac{\tau G_0}{\sqrt{2\eta}L}}\right]^2}, \tag{34}$$

*where $\eta := \min\left\{1, \frac{\tau}{\|\nabla f(x_0)\|}\right\}$, $F_0 := f(x_0) - f_{\inf}$ and $G_0 := \max\{0, \|\nabla f(x_0)\| - \tau\}$, then*

$$2L\sqrt{\gamma\phi_0} \leq \frac{\tau}{2}.$$

*Proof.* This takes a bit of effort since $\phi_0$ depends on $\gamma$ as well. In particular,

$$
2L\sqrt{\gamma\phi_0} \overset{Equation\ 26}{=} 2L\sqrt{\gamma\left(f(x_0) - f_{\inf}\right) + \frac{\gamma^2 A_\eta}{2}\left\|\nabla f(x_0) - v_0\right\|^2}
$$

$$
\overset{Equation\ 29}{=} 2L\sqrt{\gamma F_0 + \frac{\gamma^2 A_\eta}{2}G_0^2}
$$

$$
\leq 2L\sqrt{\gamma F_0 + \frac{\gamma^2}{2\eta}G_0^2},
$$

where we reach the last inequality by the fact that $A_\eta = \frac{1}{1-(1-\eta)\left(1-\frac{\eta}{2}\right)} \leq \frac{1}{1-(1-\eta)} = \frac{1}{\eta}$. By subadditivity of $t \mapsto \sqrt{t}$, we therefore get

$$
2L\sqrt{\gamma\phi_0} \overset{Equation\ 16}{\leq} 2L\sqrt{F_0}\sqrt{\gamma} + \sqrt{\frac{2}{\eta}}LG_0\gamma.
$$

Hence, any $\gamma > 0$ satisfying

$$
2L\sqrt{F_0}\sqrt{\gamma} + \sqrt{\frac{2}{\eta}}LG_0\gamma \leq \frac{\tau}{2} \tag{35}
$$

also satisfies $2L\sqrt{\gamma\phi_0} \leq \frac{\tau}{2}$. The condition Equation 35 can be re-written as

$$
\frac{1}{(\sqrt{\gamma})^2} - \frac{4}{\tau}\sqrt{F_0}\cdot L - \sqrt{\gamma}\cdot\frac{2\sqrt{2}}{\tau\sqrt{\eta}}\frac{G_0}{L}\cdot L^2 \geq 0.
$$

It remains to apply Lemma D.4 with $\beta_1 = 4\sqrt{F_0}/\tau$ and $\beta_2 = 2\sqrt{2}G_0/(\tau\sqrt{\eta}L)$.

$\square$

**Claim F.5.** *Pick any $k \geq 0$ and assume that $\|\nabla f(x_{k+1}) - v_k\| \leq \|\nabla f(x_0)\|$. Also let $\eta = \min\left\{1, \frac{\tau}{\|\nabla f(x_0)\|}\right\}$. Then*

$$
\|\nabla f(x_{k+1}) - v_{k+1}\|^2 \leq (1-\eta)\left(1 - \frac{\eta}{2}\right)\|\nabla f(x_k) - v_k\|^2 + \left(1 + \frac{2}{\eta}\right)(1-\eta)^2 L^2\|x_{k+1} - x_k\|^2, \tag{36}
$$

*and*

$$
\phi_{k+1} \leq \phi_k - \frac{\gamma}{2}\|\nabla f(x_k)\|^2 - \frac{1}{2\gamma}\left(1 - \gamma L - \gamma^2 A_\eta\left(1 + \frac{2}{\eta}\right)(1-\eta)^2 L^2\right)\|x_{k+1} - x_k\|^2. \tag{37}
$$

*If moreover the step-size satisfies*

$$
0 < \gamma \leq \frac{1}{L\left(1 + \sqrt{1 + 2A_\eta(1-\eta)^2\left(1 + \frac{2}{\eta}\right)}\right)}, \tag{38}
$$

*then*

$$
\phi_{k+1} \leq \phi_k - \frac{\gamma}{2}\|\nabla f(x_k)\|^2 - \frac{\gamma}{4}\|v_k\|^2. \tag{39}
$$

*Proof.* Since $\|\nabla f(x_{k+1}) - v_k\| \leq \|\nabla f(x_0)\|$,

$$
\eta_{k+1} \overset{Equation\ 24}{=} \min\left\{1, \frac{\tau}{\|\nabla f(x_{k+1}) - v_k\|}\right\} \geq \min\left\{1, \frac{\tau}{\|\nabla f(x_0)\|}\right\} = \eta. \tag{40}
$$

Therefore,

$$
\begin{aligned}
\|\nabla f(x_{k+1}) - v_{k+1}\|^2 \quad &\overset{Equation\ 25}{=} \quad (1 - \eta_{k+1})^2 \|\nabla f(x_{k+1}) - v_k\|^2 \\
&\overset{Equation\ 40}{\leq} \quad (1 - \eta)^2 \|\nabla f(x_{k+1}) - v_k\|^2 \\
&= \quad (1 - \eta)^2 \|\nabla f(x_k) - v_k + \nabla f(x_{k+1}) - \nabla f(x_k)\|^2 \\
&\overset{Equation\ 17}{\leq} \quad (1 + \theta)(1 - \eta)^2 \|\nabla f(x_k) - v_k\|^2 \\
&\qquad + \left(1 + \theta^{-1}\right)(1 - \eta)^2 \|\nabla f(x_{k+1}) - \nabla f(x_k)\|^2 \\
&\leq \quad (1 + \theta)(1 - \eta)^2 \|\nabla f(x_k) - v_k\|^2 \\
&\qquad + \left(1 + \theta^{-1}\right)(1 - \eta)^2 L^2 \|x_{k+1} - x_k\|^2 ,
\end{aligned}
$$

where we have the freedom to choose $\theta > 0$. To obtain Equation 36, it remains to choose $\theta = \frac{\eta}{2}$ and apply the inequality $(1 - \eta)(1 + \frac{\eta}{2}) \leq 1 - \frac{\eta}{2}$ (which holds for any $\eta \in \mathbb{R}$).

Furthermore, by combining Equation 36 with Lemma D.3, we get

$$
\begin{aligned}
\phi_{k+1} \quad &\overset{Equation\ 26}{=} \quad f(x_{k+1}) - f_{\inf} + \frac{\gamma A_\eta}{2} \|\nabla f(x_{k+1}) - v_{k+1}\|^2 \\
&\overset{Equation\ 20}{\leq} \quad f(x_k) - f_{\inf} - \frac{\gamma}{2} \|\nabla f(x_k)\|^2 - \left(\frac{1}{2\gamma} - \frac{L}{2}\right) \|x_{k+1} - x_k\|^2 + \frac{\gamma}{2} \|\nabla f(x_k) - v_k\|^2 \\
&\qquad + \frac{\gamma A_\eta}{2} \|\nabla f(x_{k+1}) - v_{k+1}\|^2 \\
&\overset{Equation\ 26}{=} \quad \phi_k - \frac{\gamma}{2} \|\nabla f(x_k)\|^2 - \left(\frac{1}{2\gamma} - \frac{L}{2}\right) \|x_{k+1} - x_k\|^2 + \left(\frac{\gamma}{2} - \frac{\gamma A_\eta}{2}\right) \|\nabla f(x_k) - v_k\|^2 \\
&\qquad + \frac{\gamma A_\eta}{2} \|\nabla f(x_{k+1}) - v_{k+1}\|^2 \\
&\overset{Equation\ 36}{\leq} \quad \phi_k - \frac{\gamma}{2} \|\nabla f(x_k)\|^2 - \left(\frac{1}{2\gamma} - \frac{L}{2} - \frac{\gamma A_\eta}{2} \left(1 + \frac{2}{\eta}\right)(1 - \eta)^2 L^2\right) \|x_{k+1} - x_k\|^2 \\
&\qquad + \left(\frac{\gamma}{2} + \frac{\gamma A_\eta}{2}(1 - \eta)\left(1 - \frac{\eta}{2}\right) - \frac{\gamma A_\eta}{2}\right) \|\nabla f(x_k) - v_k\|^2 \\
&= \quad \phi_k - \frac{\gamma}{2} \|\nabla f(x_k)\|^2 - \frac{\alpha}{2\gamma} \|x_{k+1} - x_k\|^2 ,
\end{aligned}
\tag{41}
$$

where $\alpha := 1 - \gamma L - \gamma^2 A_\eta \left(1 + \frac{2}{\eta}\right)(1 - \eta)^2 L^2$, and in the last step the term corresponding to $\|\nabla f(x_k) - v_k\|^2$ vanished because $A_\eta := \frac{1}{1 - (1-\eta)\left(1 - \frac{\eta}{2}\right)}$.

Finally, if the step-size $\gamma$ satisfies Equation 38, then from Lemma D.4 with $\beta_1 = 2$ and $\beta_2 = 2A_\eta(1 - \eta)^2 \left(1 + \frac{2}{\eta}\right)$, this condition implies $\alpha \geq \frac{1}{2}$ and thus

$$
\begin{aligned}
\phi_{k+1} \quad &\overset{Equation\ 41}{\leq} \quad \phi_k - \frac{\gamma}{2} \|\nabla f(x_k)\|^2 - \frac{1}{4\gamma} \|x_{k+1} - x_k\|^2 \\
&\overset{Equation\ 21}{=} \quad \phi_k - \frac{\gamma}{2} \|\nabla f(x_k)\|^2 - \frac{\gamma}{4} \|v_k\|^2 .
\end{aligned}
$$

$\square$

### F.2 Proof of Lemma 5.1

We will derive the descent inequality

$$
\phi_{k+1} \leq \phi_k - \frac{\gamma}{2} \|\nabla f(x_k)\|^2 .
$$

We consider two cases: (1) when $\tau < \|\nabla f(x_k) - v_{k-1}\|$ and (2) when $\|\nabla f(x_k) - v_{k-1}\| \leq \tau$.

For notational convenience, we assume that $\nabla f(x_{-1}) = v_{-1} = 0$ and $\phi_{-1} = \phi_0$.

### F.3 CASE (1): $\tau < \|\nabla f(x_k) - v_{k-1}\|$

Assume that

$$\tau < \|\nabla f(x_k) - v_{k-1}\|. \tag{42}$$

To derive the descent inequality we will show by induction the stronger result: $\|\nabla f(x_k) - v_{k-1}\| \leq B - \frac{k\tau}{2}$, where $B := \|\nabla f(x_0)\|$ and

$$\phi_k \leq \phi_{k-1} - \frac{\gamma}{2}\|\nabla f(x_{k-1})\|^2 - \frac{\gamma}{4}\|v_{k-1}\|^2 \tag{43}$$

for any $k \geq 0$. The base of the induction is trivial: when $k = 0$ we have $\|\nabla f(x_k) - v_{k-1}\| = \|\nabla f(x_0) - v_{-1}\| = \|\nabla f(x_0)\| := B > \tau$ and Equation 43 holds by definition. Next, we assume that for some $k \geq 0$ inequalities $\|\nabla f(x_t) - v_{t-1}\| \leq B$ and Equation 43 hold for $t = 0, 1, \ldots, k$. The only task is to show these conditions hold for $k + 1$.

If $\gamma \leq 1/L$, then

$$\|\nabla f(x_{k-1})\|^2 \overset{\underset{Equation\ 18}{}}{\leq} 2L[f(x_{k-1}) - f_{\inf}]$$
$$\overset{\underset{Equation\ 26}{}}{\leq} 2L\phi_{k-1}$$
$$\overset{\underset{Equation\ 43}{}}{\leq} 2L\phi_0$$
$$\leq \frac{2}{\gamma}\phi_0.$$

If $\gamma \leq \frac{1 - \frac{1}{\sqrt{2}}}{L}$ and $v_{-1} = 0$, then by the fact that $\|v_{k-1}\| \leq \sqrt{(4/\gamma)\phi_0}$ (from Equation 43) and by Claim F.3 we have for $k \geq 0$

$$\|v_k\| \leq \sqrt{\frac{4}{\gamma}\phi_0}.$$

Next, from Claim F.2 with Equation 42,

$$\begin{aligned}
\|\nabla f(x_{k+1}) - v_k\| &\leq \|\nabla f(x_k) - v_{k-1}\| - \tau + L\gamma\|v_k\| \\
&\leq \|\nabla f(x_k) - v_{k-1}\| - \tau + 2L\sqrt{\gamma\phi_0}.
\end{aligned}$$

**STEP: Small stepsize.**

If $\gamma > 0$ satisfies Equation 34, then from Claim F.4 we have $2L\sqrt{\gamma\phi_0} \leq \frac{\tau}{2}$. Hence, the above inequality and the inductive assumption imply

$$\|\nabla f(x_{k+1}) - v_k\| \leq \|\nabla f(x_k) - v_{k-1}\| - \frac{\tau}{2} \leq \cdots \leq \|\nabla f(x_0)\| - \frac{(k+1)\tau}{2}. \tag{44}$$

In conclusion, Eq. Equation 44 implies that $\|\nabla f(x_k) - v_{k-1}\| \leq \|\nabla f(x_0)\| := B$ for any $k \geq 0$.

**STEP: Descent inequality.**

If the step-size $\gamma > 0$ satisfies Equation 38, then from Claim F.5

$$\phi_{k+1} \leq \phi_k - \frac{\gamma}{2}\|\nabla f(x_k)\|^2 - \frac{\gamma}{4}\|v_k\|^2.$$

This concludes the proof in case (1).

### F.4 CASE (2): $\|\nabla f(x_k) - v_{k-1}\| \leq \tau$

We show by the induction that $\|\nabla f(x_k) - v_{k-1}\| \leq \tau$ and that

$$\phi_k \leq \phi_{k-1} - \frac{\gamma}{2}\|\nabla f(x_{k-1})\|^2 \tag{45}$$

for any $k \geq 0$. First the base case at $k = 0$ is trivial, i.e. $\|\nabla f(x_0)\| := \bar{B} \leq \tau$ and $\phi_0 = \phi_{-1}$. Next, assume $\|\nabla f(x_t) - v_{t-1}\| \leq \tau$ and Equation 45 holds for $t = 0, 1, \ldots, k$. Then, we show that these

conditions hold for $k + 1$. By the fact that $\|\nabla f(x_k) - v_{k-1}\| \leq \tau$, by Equation 23 and by the fact that $\eta_k = \eta = 1$,

$$v_k = \nabla f(x_k). \tag{46}$$

Therefore, single-node Clip21-GD described in Algorithm 2 with $n = 1$ reduces to classical gradient descent at step $k$. From the definition of $\phi_k$ and Lemma D.3,

$$
\begin{aligned}
\phi_{k+1} &= f(x_{k+1}) - f_{\inf} \\
&\overset{Equation\ 20}{\leq} f(x_k) - f_{\inf} - \frac{\gamma}{2} \|\nabla f(x_k)\|^2 - \left(\frac{1}{2\gamma} - \frac{L}{2}\right) \|x_{k+1} - x_k\|^2 + \frac{\gamma}{2} \|\nabla f(x_k) - v_k\|^2 \\
&\leq \phi_k - \frac{\gamma}{2} \|\nabla f(x_k)\|^2 - \left(\frac{1}{2\gamma} - \frac{L}{2}\right) \|x_{k+1} - x_k\|^2.
\end{aligned}
$$

If $\gamma \leq 1/L$, then we get Equation 45, $0 \leq \phi_k \leq \phi_0$ and

$$\|\nabla f(x_k)\| \leq \sqrt{\|\nabla f(x_k)\|^2} \leq \sqrt{\frac{2}{\gamma}[\phi_k - \phi_{k+1}]} \leq \sqrt{\frac{2}{\gamma}\phi_k} \overset{Equation\ 45}{\leq} \sqrt{\frac{2}{\gamma}\phi_0}.$$

We complete the proof by showing that $\|\nabla f(x_k) - v_{k-1}\| \leq \tau$ implies $\|\nabla f(x_{k+1}) - v_k\| \leq \tau$. In this case,

$$
\begin{aligned}
\|\nabla f(x_{k+1}) - v_k\| &\leq L \|x_{k+1} - x_k\| \\
&\overset{Equation\ 46}{=} L\gamma \|\nabla f(x_k)\| \\
&\overset{Equation\ 15}{\leq} L\gamma \|\nabla f(x_k) - v_{k-1}\| + L\gamma \|v_{k-1}\| \\
&\leq L\gamma\tau + L\gamma \|v_{k-1}\|.
\end{aligned}
$$

From Claim F.3 (clipping active at $k - 1$) and from Equation 45 (clipping inactive at $k - 1$)

$$\|v_{k-1}\| \leq \max\left(\sqrt{\frac{4}{\gamma}\phi_0}, \sqrt{\frac{2}{\gamma}\phi_0}\right) \leq \sqrt{\frac{4}{\gamma}\phi_0}.$$

Therefore,

$$\|\nabla f(x_{k+1}) - v_k\| \leq L\gamma\tau + 2L\sqrt{\gamma\phi_0}. \tag{47}$$

In conclusion, $\|\nabla f(x_k) - v_{k-1}\| \leq \tau$ implies $\|\nabla f(x_{k+1}) - v_k\| \leq \tau$ if $\gamma$ satisfies

$$L\gamma\tau \leq \frac{\tau}{2} \quad \text{and} \quad 2L\sqrt{\gamma\phi_0} \leq \frac{\tau}{2}. \tag{48}$$

From Claim F.4, we can express step-size conditions Equation 48 equivalently as:

$$\gamma \leq \frac{1}{2L} \quad \text{and} \quad \gamma \leq \frac{\tau^2}{4L^2\left[\sqrt{F_0} + \sqrt{F_0 + \frac{\tau G_0}{\sqrt{2}\eta L}}\right]^2},$$

where $\eta = \frac{\tau}{\|\nabla f(x_0)\|}$, $F_0 = f(x_0) - f_{\inf}$ and $G_0 = |\|\nabla f(x_0)\| - \tau|$. Putting all the conditions on $\gamma$ together, we obtain the results.

## G  PROOF OF PROPOSITION 5.2

If $\|\nabla f(x_0)\| \leq \tau$, then from Equation 47 and step-size condition from Theorem 5.3, we prove that the clipping operator is always turned off for all $k \geq 0$, i.e., $\|\nabla f(x_k) - v_{k-1}\| \leq \tau$ implying that $v_k = \nabla f(x_k)$ for all $k \geq 0$.

If $\|\nabla f(x_0)\| > \tau$, then the clipping operator is turned on at the beginning. Moreover, for all $k \geq 0$ such that $\|\nabla f(x_k) - v_{k-1}\| > \tau$ (clipping is turned on), we have from the derivation of Equation 44 and the step-size condition of Theorem 5.3

$$\|\nabla f(x_k) - v_{k-1}\| \leq \|\nabla f(x_{k-1}) - v_{k-2}\| - \frac{\tau}{2} \leq \ldots \leq \|\nabla f(x_0)\| - k\frac{\tau}{2}.$$

Therefore, the situation when $\tau < \|\nabla f(x_k) - v_{k-1}\|$ is possible only for $0 \leq k < k^\star$ with $k^\star = \frac{2}{\tau}(\|\nabla f(x_0)\| - \tau) + 1$. After that, the clipping operator always turns off, i.e., $\|\bar{\nabla} f(x_k) - v_{k-1}\| \leq \tau$ for $k \geq k^\star$.

## H   PROOF OF THEOREM 5.3

Let $\hat{x}_K$ be selected uniformly at random from $\{x_0, x_1, \ldots, x_{K-1}\}$. Then,

$$\mathbf{E}\left[\|\nabla f(\hat{x}_K)\|^2\right] = \frac{1}{K}\sum_{k=0}^{K-1}\|\nabla f(x_k)\|^2.$$

From Theorem 5.3,

$$\mathbf{E}\left[\|\nabla f(\hat{x}_K)\|^2\right] \le \frac{2}{\gamma}\frac{1}{K}\sum_{k=0}^{K-1}(\phi_k - \phi_{k+1}) = \frac{2(\phi_0 - \phi_K)}{\gamma K} \le \frac{2\phi_0}{\gamma K}.$$

Next, we consider the case when

$$\gamma = \min\left(\frac{1 - 1/\sqrt{2}}{L}, \frac{1}{L(1 + \sqrt{1 + 2\beta_1})}, \frac{\tau^2}{4L^2\left[\sqrt{F_0} + \sqrt{\beta_2}\right]^2}\right).$$

First, we have

$$\beta_1 = \frac{(1-\eta)^2(1 + {}^2\!/\!\eta)}{1 - (1-\eta)(1 - {}^2\!/\!\eta)} \le \frac{1 + {}^2\!/\!\eta}{1 - (1-\eta)} = \frac{2}{\eta^2} + \frac{1}{\eta} = \mathcal{O}\left(\frac{1}{\eta^2}\right) = \mathcal{O}\left(1 + \frac{\|\nabla f(x_0)\|^2}{\tau^2}\right),$$

$$\beta_2 = F_0 + \frac{\tau G_0}{\sqrt{2\eta}L} \le F_0 + \frac{\tau\max(\|\nabla f(x_0)\|, \tau)}{\sqrt{2\max\left(1, \frac{\tau}{\|\nabla f(x_0)\|}\right)}L} = F_0 + \frac{\tau\|\nabla f(x_0)\|\sqrt{\max\left(1, \frac{\tau}{\|\nabla f(x_0)\|}\right)}}{\sqrt{2}L}$$

$$= \mathcal{O}\left(F_0 + \frac{\tau\|\nabla f(x_0)\|}{L} + \frac{\tau^{3/2}\sqrt{\|\nabla f(x_0)\|}}{L}\right).$$

Using this, we estimate $^1\!/\!\gamma$ as

$$\frac{1}{\gamma} = \max\left(\frac{L}{1 - 1/\sqrt{2}}, L(1 + \sqrt{1 + 2\beta_1}), \frac{4L^2\left[\sqrt{F_0} + \sqrt{\beta_2}\right]^2}{\tau^2}\right)$$

$$= \mathcal{O}\left(L\left(1 + \frac{\|\nabla f(x_0)\|}{\tau}\right) + \frac{L^2 F_0}{\tau^2} + \frac{L\|\nabla f(x_0)\|}{\tau} + \frac{L\sqrt{\|\nabla f(x_0)\|}}{\sqrt{\tau}}\right)$$

$$= \mathcal{O}\left(L\left(1 + \frac{\|\nabla f(x_0)\|}{\tau}\right) + \frac{L^2 F_0}{\tau^2}\right).$$

Therefore, since

$$\frac{A}{\gamma} = \frac{1}{2[1 - (1-\eta)(1 - \eta/2)]} = \mathcal{O}\left(\frac{1}{\eta}\right) = \mathcal{O}\left(1 + \frac{\|\nabla f(x_0)\|}{\tau}\right)$$

and

$$\|\nabla f(x_0) - v_{-1}\|^2 = \|\nabla f(x_0)\|^2,$$

we have

$$\mathbf{E}\left[\|\nabla f(\hat{x}_K)\|^2\right] \le \frac{2\phi_0}{\gamma K} = \frac{2\left(f(x_0) - f_{\inf} + A\|\nabla f(x_0) - v_{-1}\|^2\right)}{\gamma K}$$

$$= \mathcal{O}\left(\frac{\left(1 + \frac{\|\nabla f(x_0)\|}{\tau}\right)\max(LF_0, \|\nabla f(x_0)\|^2) + \frac{L^2(F_0)^2}{\tau^2}}{K}\right),$$

which concludes the proof.

## I  MULTI-NODE Clip21-GD

In this section we analyze the convergence for multi-node Clip21-GD described in Algorithm 2. Its update can be expressed as Equation 21, where $v_k = \frac{1}{n}\sum_{i=1}^{n} v_k^i$ and

$$v_k^i = (1 - \eta_k^i)v_{k-1}^i + \eta_k^i \nabla f_i(x_k). \tag{49}$$

Here, $\eta_k^i = \min\left(1, \frac{\tau}{\|\nabla f_i(x_k) - v_{k-1}^i\|}\right)$. To facilitate our analysis, denote $\mathcal{I}_k$ as the subset from $\{1, 2, \ldots, n\}$ such that $\|\nabla f_i(x_k) - v_{k-1}^i\| > \tau$. Recall that the Lyapunov function is

$$\phi_k = f(x_k) - f^{\inf} + A\frac{1}{n}\sum_{i=1}^{n} \left\|\nabla f_i(x_k) - v_k^i\right\|^2, \tag{50}$$

where $A = \frac{\gamma}{2[1-(1-\eta)(1-\eta/2)]}$.

### I.1  CLAIMS

We first establish several simple but helpful results.

**Claim I.1.** *Let each $f_i$ have $L_i$-Lipschitz gradient. Then, for $k \geq 0$,*

$$\left\|\nabla f_i(x_{k+1}) - v_k^i\right\| \leq \max\left\{0, \left\|\nabla f_i(x_k) - v_{k-1}^i\right\| - \tau\right\} + L_{\max}\gamma\|v_k\|. \tag{51}$$

*Proof.* From the definition of the Euclidean norm,

$$
\begin{aligned}
\left\|\nabla f_i(x_{k+1}) - v_k^i\right\| &\overset{\text{Equation 15}}{\leq} \left\|\nabla f_i(x_k) - v_k^i\right\| + \left\|\nabla f_i(x_{k+1}) - \nabla f_i(x_k)\right\| \\
&\overset{\text{Equation 49}}{=} \left\|\nabla f_i(x_k) - v_{k-1}^i - \mathsf{clip}_\tau(\nabla f_i(x_k) - v_{k-1}^i)\right\| \\
&\quad + \left\|\nabla f_i(x_{k+1}) - \nabla f_i(x_k)\right\| \\
&\overset{\text{(Lemma 4.1(ii)-(iii))}}{\leq} \max\{0, \left\|\nabla f_i(x_k) - v_{k-1}^i\right\| - \tau\} + \left\|\nabla f_i(x_{k+1}) - \nabla f_i(x_k)\right\| \\
&\overset{\text{Equation 3}}{\leq} \max\{0, \left\|\nabla f_i(x_k) - v_{k-1}^i\right\| - \tau\} + L_{\max}\|x_{k+1} - x_k\| \\
&\overset{\text{Equation 21}}{=} \max\{0, \left\|\nabla f_i(x_k) - v_{k-1}^i\right\| - \tau\} + L_{\max}\gamma\|v_k\|.
\end{aligned}
$$

$\square$

**Claim I.2.** *Let $v_{-1}^i = 0$ for all $i$, $\max_i \|\nabla f_i(x_0)\| := B > \tau$ and $\gamma \leq 2/L$. Then,*

$$\|v_0\| \leq \sqrt{\frac{4}{\gamma}\phi_0} + 2(B - \tau). \tag{52}$$

*Proof.* By the fact that $v_0 = \frac{1}{n}\sum_{i=1}^{n} v_0^i = \frac{1}{n}\sum_{i=1}^{n} \mathsf{clip}_\tau(\nabla f_i(x_0))$,

$$
\begin{aligned}
\|v_0\| &\overset{\text{Equation 15}}{\leq} \|\nabla f(x_0)\| + \left\|\frac{1}{n}\sum_{i=1}^{n} \mathsf{clip}_\tau(\nabla f_i(x_0)) - \nabla f(x_0)\right\| \\
&\overset{\text{Equation 15}}{\leq} \|\nabla f(x_0)\| + \frac{1}{n}\sum_{i=1}^{n} \|\mathsf{clip}_\tau(\nabla f_i(x_0)) - \nabla f_i(x_0)\| \\
&\overset{\text{(Lemma 4.1(ii)-(iii))}}{\leq} \|\nabla f(x_0)\| + \frac{1}{n}\sum_{i=1}^{n} \max\{0, \|\nabla f_i(x_0)\| - \tau\}.
\end{aligned}
$$

If $\max_i \|\nabla f_i(x_0)\| := B > \tau$ and $\gamma \leq 2/L$, then

$$
\begin{array}{rcl}
\|v_0\| & \leq & \|\nabla f(x_0)\| + B - \tau \\
& \leq & \|\nabla f(x_0)\| + 2(B - \tau) \\
& \overset{Equation\ 18}{\leq} & \sqrt{2L[f(x_0) - f^{\inf}]} + 2(B - \tau) \\
& \overset{Equation\ 50}{\leq} & \sqrt{2L\phi_0} + 2(B - \tau) \\
& \leq & \sqrt{\frac{4}{\gamma}\phi_0} + 2(B - \tau).
\end{array}
$$

$\square$

**Claim I.3.** *Fix $k \geq 1$. Let $f$ have $L$-Lipschitz gradient. Also suppose that $\left\|\nabla f_i(x_k) - v_{k-1}^i\right\| \leq \max_i \|\nabla f_i(x_0)\| := B > \tau$ for $i \in \mathcal{I}_k$, $\|v_{k-1}\| \leq \sqrt{\frac{4}{\gamma}\phi_0} + 2(B - \tau)$, $\|\nabla f(x_{k-1})\| \leq \sqrt{\frac{2}{\gamma}\phi_0}$, and $\gamma \leq \frac{1 - \frac{1}{\sqrt{2}}}{L}$. Then,*

$$
\|v_k\| \quad \leq \quad \sqrt{\frac{4}{\gamma}\phi_0} + 2(B - \tau). \tag{53}
$$

*Proof.* From the definition of the Euclidean norm and by the fact that $v_k = \frac{1}{n}\sum_{i=1}^n v_k^i$,

$$
\begin{array}{rcl}
\|v_k\| \quad \overset{Equation\ 49}{=} & & \left\|\frac{1}{n}\sum_{i=1}^n v_{k-1}^i + \mathsf{clip}_\tau(\nabla f_i(x_k) - v_{k-1}^i)\right\| \\[3mm]
= & & \left\|\frac{1}{n}\sum_{i=1}^n \nabla f_i(x_k) + [\mathsf{clip}_\tau(\nabla f_i(x_k) - v_{k-1}^i) - (\nabla f_i(x_k) - v_{k-1}^i)]\right\| \\[3mm]
\overset{Equation\ 15}{\leq} & & \|\nabla f(x_k)\| + \frac{1}{n}\sum_{i=1}^n \left\|\mathsf{clip}_\tau(\nabla f_i(x_k) - v_{k-1}^i) - (\nabla f_i(x_k) - v_{k-1}^i)\right\| \\[3mm]
\leq & & \|\nabla f(x_k)\| + \frac{1}{n}\sum_{i=1}^n \max\{0, \left\|\nabla f_i(x_k) - v_{k-1}^i\right\| - \tau\}.
\end{array}
$$

If $\left\|\nabla f_i(x_k) - v_{k-1}^i\right\| \leq \max_i \|\nabla f_i(x_0)\| := B > \tau$ for $i \in \mathcal{I}_k$, then

$$
\begin{array}{rcl}
\|v_k\| & \leq & \|\nabla f(x_k)\| + B - \tau \\
& \overset{Equation\ 15}{\leq} & \|\nabla f(x_{k-1})\| + \|\nabla f(x_k) - \nabla f(x_{k-1})\| + B - \tau \\
& \overset{Equation\ 3}{\leq} & \|\nabla f(x_{k-1})\| + L\|x_k - x_{k-1}\| + B - \tau \\
& \overset{Equation\ 21}{=} & \|\nabla f(x_{k-1}))\| + L\gamma\|v_{k-1}\| + B - \tau.
\end{array}
$$

If $\|v_{k-1}\| \leq \sqrt{\frac{4}{\gamma}\phi_0} + 2(B - \tau)$ and $\|\nabla f(x_{k-1})\| \leq \sqrt{\frac{2}{\gamma}\phi_0}$, then

$$
\|v_k\| \quad \leq \quad (L\gamma + 1/\sqrt{2})\sqrt{\frac{4}{\gamma}\phi_0} + (2L\gamma + 1)(B - \tau).
$$

If $\gamma \leq \frac{1 - \frac{1}{\sqrt{2}}}{L}$, then $\gamma \leq 1/(2L)$ and

$$
\|v_k\| \quad \leq \quad \sqrt{\frac{4}{\gamma}\phi_0} + 2(B - \tau).
$$

$\square$

**Claim I.4.** *If*

$$0 < \gamma \le \frac{\tau^2}{16L_{\max}^2(\sqrt{F_0} + \sqrt{F_0 + \frac{G_0\tau}{2\sqrt{2\eta}L_{\max}}})^2}, \tag{54}$$

*where* $\eta := \min\left\{1, \frac{\tau}{\max_i\|\nabla f_i(x_0)\|}\right\}$, $F_0 := f(x_0) - f_{\inf}$ *and* $G_0 := \sqrt{\frac{1}{n}\sum_{i=1}^n(\|\nabla f_i(x_0)\| - \tau)^2}$, *then*

$$4L_{\max}\sqrt{\gamma\phi_0} \le \tau/2. \tag{55}$$

*Proof.* By the definition of $\phi_0$ and by the fact that $A = \frac{\gamma}{2[1-(1-\eta)(1-\eta/2)]} \le \frac{\gamma}{2[1-(1-\eta)]} = \frac{\gamma}{2\eta}$,

$$
\begin{aligned}
2L_{\max}\sqrt{\gamma\phi_0} &= 2L_{\max}\sqrt{\gamma[f(x_0) - f_{\inf}] + \gamma A\frac{1}{n}\sum_{i=1}^n\left\|\nabla f_i(x_0) - v_0^i\right\|^2} \\
&\le 2L_{\max}\sqrt{\gamma[f(x_0) - f_{\inf}] + \frac{\gamma^2}{2\eta}\frac{1}{n}\sum_{i=1}^n\left\|\nabla f_i(x_0) - v_0^i\right\|^2}.
\end{aligned}
$$

Since $v_{-1}^i = 0$ for all $i$, we have $v_0^i = \mathsf{clip}_\tau(\nabla f_i(x_0))$ and

$$2L_{\max}\sqrt{\gamma\phi_0} \overset{\text{(Lemma 4.1(ii)-(iii))}}{\le} 2L_{\max}\sqrt{\gamma[f(x_0) - f_{\inf}] + \frac{\gamma^2}{2\eta}\frac{1}{n}\sum_{i=1}^n\max\{0, \|\nabla f_i(x_0)\| - \tau\}^2}$$

$$\overset{Equation\ 16}{\le} 2L_{\max}\sqrt{\gamma}\sqrt{F_0} + \gamma L_{\max}\sqrt{\frac{2}{\eta}}G_0,$$

where $F_0 := f(x_0) - f_{\inf}$ and $G_0 := \sqrt{\frac{1}{n}\sum_{i=1}^n(\|\nabla f_i(x_0)\| - \tau)^2}$.

Hence, any $\gamma > 0$ satisfying

$$4L_{\max}\sqrt{\gamma}\sqrt{F_0} + \gamma 2L_{\max}\sqrt{\frac{2}{\eta}}G_0 \le \tau/2 \tag{56}$$

also satisfies $4L_{\max}\sqrt{\gamma\phi_0} \le B/2$. Condition Equation 56 can be rewritten as:

$$\frac{1}{\sqrt{\gamma}} - \frac{8\sqrt{F_0}}{\tau}L_{\max} - \sqrt{\gamma}\frac{4\sqrt{2}}{\sqrt{\eta}}\frac{G_0}{L_{\max}\tau}L_{\max}^2 \ge 0.$$

Finally, by Lemma D.4 with $L = L_{\max}$, $\beta_1 = \frac{8\sqrt{F_0}}{\tau}$ and $\beta_2 = \frac{4\sqrt{2}}{\sqrt{\eta}}\frac{G_0}{L_{\max}\tau}$, we have

$$0 < \sqrt{\gamma} \le \frac{\tau}{4L_{\max}(\sqrt{F_0} + \sqrt{F_0 + \frac{G_0\tau}{2\sqrt{2\eta}L_{\max}}})}.$$

By taking the square, we complete our proof. $\qquad\square$

### I.2 PROOF OF THEOREM 5.6

We then derive the descent inequality

$$\phi_{k+1} \le \phi_k - \frac{\gamma}{2}\|\nabla f(x_k)\|^2,$$

for two possible cases: (1) when $|\mathcal{I}_k| > 0$ and (2) when $|\mathcal{I}_k| = 0$. For notational convenience we assume that $\nabla f_i(x_{-1}) = v_{-1}^i = 0$ and $\phi_{-1} = \phi_0$.

### I.3 CASE (1): $|\mathcal{I}_k| > 0$

Suppose $|\mathcal{I}_k| > 0$. To derive the descent inequality we show by induction the stronger result: $\left\| \nabla f_i(x_k) - v_{k-1}^i \right\| \leq B - \frac{k\tau}{2}$, where $B := \max_i \|\nabla f_i(x_0)\|$ and

$$\phi_k \leq \phi_{k-1} - \frac{\gamma}{2} \|\nabla f(x_{k-1})\|^2 - \frac{\gamma}{4} \|v_{k-1}\|^2 \tag{57}$$

for any $k \geq 0$. First when $k = 0$ we have $\left\| \nabla f_i(x_0) - v_{-1}^i \right\| = \|\nabla f_i(x_0)\| \leq \max_i \|\nabla f_i(x_0)\| = B$ where $B > \tau$ and Equation 57 holds by definition. Next, we assume that $\left\| \nabla f_i(x_t) - v_{t-1}^i \right\| \leq B$ and Equation 57 hold for $t = 0, 1, \ldots, k$.

If $\gamma \leq 1/L$, then

$$
\|\nabla f(x_{k-1})\|^2 \overset{\substack{Equation\ 18}}{\leq} 2L[f(x_{k-1}) - f_{\inf}]
$$
$$
\overset{\substack{Equation\ 26}}{\leq} 2L\phi_{k-1}
$$
$$
\overset{\substack{Equation\ 57}}{\leq} 2L\phi_0
$$
$$
\leq \frac{2}{\gamma}\phi_0.
$$

If $\gamma \leq (1 - 1/\sqrt{2})/L$, then by using the above inequality, and also Claim I.2 and I.3, for $k \geq 0$

$$\|v_k\| \leq \sqrt{\frac{4}{\gamma}\phi_0} + 2(B - \tau).$$

Next, from the above inequality and from Claim I.1,

$$
\begin{aligned}
\left\| \nabla f_i(x_{k+1}) - v_k^i \right\| &\leq \left\| \nabla f_i(x_k) - v_{k-1}^i \right\| - \tau + L_{\max}\gamma \|v_k\| \\
&\leq \left\| \nabla f_i(x_k) - v_{k-1}^i \right\| - \tau + L_{\max}\gamma\sqrt{\frac{4}{\gamma}\phi_0} + 2L_{\max}\gamma(B - \tau) \\
&= \left\| \nabla f_i(x_k) - v_{k-1}^i \right\| - \tau + 2L_{\max}\sqrt{\gamma\phi_0} + 2L_{\max}\gamma(B - \tau).
\end{aligned}
$$

**STEP: Small step-size**

The above inequality and the inductive assumption imply

$$\left\| \nabla f_i(x_{k+1}) - v_k^i \right\| \leq \left\| \nabla f_i(x_k) - v_{k-1}^i \right\| - \frac{\tau}{2} \leq B - \frac{(k+1)\tau}{2} \tag{58}$$

if the step-size $\gamma > 0$ satisfies

$$2L_{\max}\gamma B \leq \tau/4 \quad \text{and} \quad 4L_{\max}\sqrt{\gamma\phi_0} \leq \tau/2.$$

By Claim I.4, this condition can be expressed equivalently as:

$$\gamma \leq \frac{\tau}{8L_{\max}B} \quad \text{and} \quad \gamma \leq \frac{\tau^2}{16L_{\max}^2(\sqrt{F_0} + \sqrt{F_0 + \frac{G_0\tau}{2\sqrt{2\eta}L_{\max}}})^2}.$$

**STEP: Descent inequality**

It remains to prove the descent inequality. By the inductive assumption proved above, we then have for $i \in \mathcal{I}_{k+1}$,

$$\eta_{k+1}^i = \frac{\tau}{\left\| \nabla f_i(x_{k+1}) - v_k^i \right\|} \geq \frac{\tau}{B} = \eta. \tag{59}$$

Therefore,

$$
\left\|\nabla f_i(x_{k+1}) - v^i_{k+1}\right\|^2 \overset{Equation\ 49}{=} 0 \cdot \mathbb{1}(i \in \mathcal{I}'_{k+1}) + (1 - \eta^i_{k+1})^2 \left\|\nabla f_i(x_{k+1}) - v^i_k\right\|^2 \cdot \mathbb{1}(i \in \mathcal{I}_{k+1})
$$

$$
\overset{Equation\ 59}{\leq} 0 \cdot \mathbb{1}(i \in \mathcal{I}'_{k+1}) + (1 - \eta)^2 \left\|\nabla f_i(x_{k+1}) - v^i_k\right\|^2 \cdot \mathbb{1}(i \in \mathcal{I}_{k+1})
$$

$$
\leq (1 - \eta)^2 \left\|\nabla f_i(x_{k+1}) - v^i_k\right\|^2
$$

$$
\overset{Equation\ 17}{\leq} (1 + \theta)(1 - \eta)^2 \left\|\nabla f_i(x_k) - v^i_k\right\|^2
$$
$$
+ (1 + 1/\theta)(1 - \eta)^2 \left\|\nabla f_i(x_{k+1}) - \nabla f_i(x_k)\right\|^2
$$

$$
\overset{Equation\ 2}{\leq} (1 + \theta)(1 - \eta)^2 \left\|\nabla f_i(x_k) - v^i_k\right\|^2
$$
$$
+ (1 + 1/\theta)(1 - \eta)^2 L^2_{\max} \left\|x_{k+1} - x_k\right\|^2
$$

where $\theta > 0$. Taking $\theta = \eta/2$ and applying inequality $(1 - \eta)(1 + \eta/2) \leq 1 - \eta/2$, we get

$$
\left\|\nabla f_i(x_{k+1}) - v^i_{k+1}\right\|^2 \leq (1 - \eta)(1 - \eta/2) \left\|\nabla f_i(x_k) - v^i_k\right\|^2
$$
$$
+ (1 + 2/\eta)(1 - \eta)^2 L^2_{\max} \left\|x_{k+1} - x_k\right\|^2. \tag{60}
$$

Next, we combine the above inequality with Lemma D.3:

$$
\phi_{k+1} = f(x_{k+1}) - f_{\inf} + A \frac{1}{n} \sum_{i=1}^n \left\|\nabla f_i(x_{k+1}) - v^i_{k+1}\right\|^2
$$

$$
\overset{Equation\ 20}{\leq} f(x_k) - f_{\inf} - \frac{\gamma}{2} \left\|\nabla f(x_k)\right\|^2 - \left(\frac{1}{2\gamma} - \frac{L}{2}\right) \left\|x_{k+1} - x_k\right\|^2
$$

$$
+ \frac{\gamma}{2} \frac{1}{n} \sum_{i=1}^n \left\|\nabla f_i(x_k) - v^i_k\right\|^2 + A \frac{1}{n} \sum_{i=1}^n \left\|\nabla f_i(x_{k+1}) - v^i_{k+1}\right\|^2
$$

$$
= \phi_k - \frac{\gamma}{2} \left\|\nabla f(x_k)\right\|^2 - \left(\frac{1}{2\gamma} - \frac{L}{2}\right) \left\|x_{k+1} - x_k\right\|^2
$$

$$
+ \left(\frac{\gamma}{2} - A\right) \frac{1}{n} \sum_{i=1}^n \left\|\nabla f_i(x_k) - v^i_k\right\|^2 + A \frac{1}{n} \sum_{i=1}^n \left\|\nabla f_i(x_{k+1}) - v^i_{k+1}\right\|^2
$$

$$
\overset{Equation\ 60}{\leq} \phi_k - \frac{\gamma}{2} \left\|\nabla f(x_k)\right\|^2 - \left(\frac{1}{2\gamma} - \frac{L}{2} - A\left(1 + \frac{2}{\eta}\right)(1 - \eta)^2(L_{\max})^2\right) \left\|x_{k+1} - x_k\right\|^2
$$

$$
+ \left(\frac{\gamma}{2} + A(1 - \eta)(1 - \eta/2) - A\right) \frac{1}{n} \sum_{i=1}^n \left\|\nabla f_i(x_k) - v^i_k\right\|^2.
$$

Since $A = \frac{\gamma}{2[1 - (1 - \eta)(1 - \eta/2)]}$, we get

$$
\phi_{k+1} \leq \phi_k - \frac{\gamma}{2} \left\|\nabla f(x_k)\right\|^2 - \frac{1}{2\gamma} \left(1 - \gamma L - \gamma^2 \cdot \frac{(1 + 2/\eta)(1 - \eta)^2}{[1 - (1 - \eta)(1 - \eta/2)]}(L_{\max})^2\right) \left\|x_{k+1} - x_k\right\|^2.
$$

If the step-size $\gamma$ satisfies

$$
0 < \gamma \leq \frac{1}{L\left(1 + \sqrt{1 + 2\frac{(1 - \eta)^2(1 + 2/\eta)}{[1 - (1 - \eta)(1 - \eta/2)]}\left(\frac{L_{\max}}{L}\right)^2}\right)},
$$

then from Lemma D.4 with $L = 1$, $\beta_1 = 2L$ and $\beta_2 = 2\frac{(1 - \eta)^2(1 + 2/\eta)}{[1 - (1 - \eta)(1 - \eta/2)]}(L_{\max})^2$, this condition implies $1 - \gamma L - \gamma^2 \cdot \frac{(1 - \eta)^2(1 + 2/\eta)}{[1 - (1 - \eta)(1 - \eta/2)]}(L_{\max})^2 \geq \frac{1}{2}$ and thus

$$
\phi_{k+1} \leq \phi_k - \frac{\gamma}{2} \left\|\nabla f(x_k)\right\|^2 - \frac{1}{4\gamma} \left\|x_{k+1} - x_k\right\|^2.
$$

Since $x_{k+1} - x_k = -\gamma v_k$,

$$
\phi_{k+1} \leq \phi_k - \frac{\gamma}{2} \left\|\nabla f(x_k)\right\|^2 - \frac{\gamma}{4} \left\|v_k\right\|^2. \tag{61}
$$

This concludes the proof in the case (1).

### I.4 CASE (2): $|\mathcal{I}_k| = 0$

Suppose $|\mathcal{I}_k| = 0$. We show by induction that the following holds: for any $k \geq 0$, $\left\| \nabla f_i(x_k) - v_{k-1}^i \right\| \leq \tau$ for all $i$ and

$$\phi_k \leq \phi_{k-1} - \frac{\gamma}{2} \left\| \nabla f(x_{k-1}) \right\|^2. \tag{62}$$

First the base case is trivial, i.e. $\|\nabla f(x_0)\| := \bar{B} \leq \tau$ and $\phi_0 = \phi_{-1}$. Next, suppose that $\left\| \nabla f_i(x_t) - v_{t-1}^i \right\| \leq \tau$ for all $i$, and Equation 62 holds for $t = 0, 1, \ldots, k$. We now show that these conditions hold for $k + 1$. Since $\left\| \nabla f_i(x_k) - v_{k-1}^i \right\| \leq \tau$ for all $i$, we have $v_k^i = \nabla f_i(x_k)$. Therefore, Clip21-GD described in Algorithm 2 reduces to classical gradient descent at step $k$. From the definition of $\phi_k$ and Lemma D.3,

$$\phi_{k+1} = f(x_{k+1}) - f_{\inf}$$

$$\overset{Equation\ 20}{\leq} f(x_k) - f_{\inf} - \frac{\gamma}{2} \left\| \nabla f(x_k) \right\|^2 - \left( \frac{1}{2\gamma} - \frac{L}{2} \right) \left\| x_{k+1} - x_k \right\|^2 + \frac{\gamma}{2} \left\| \nabla f(x_k) - v_k \right\|^2$$

$$\leq \phi_k - \frac{\gamma}{2} \left\| \nabla f(x_k) \right\|^2 - \left( \frac{1}{2\gamma} - \frac{L}{2} \right) \left\| x_{k+1} - x_k \right\|^2.$$

If $\gamma \leq 1/L$, then Equation 62 holds, $0 \leq \phi_k \leq \phi_0$ and

$$\|\nabla f(x_k)\| \leq \sqrt{\|\nabla f(x_k)\|^2} \leq \sqrt{\frac{2}{\gamma} [\phi_k - \phi_{k+1}]} \leq \sqrt{\frac{2}{\gamma} \phi_k} \leq \sqrt{\frac{2}{\gamma} \phi_0}.$$

Finally, we show that $\left\| \nabla f_i(x_k) - v_{k-1}^i \right\| \leq \tau$ for all $i$ implies $\left\| \nabla f_i(x_{k+1}) - v_k^i \right\| \leq \tau$ for all $i$. Indeed, in this case, we have $v_k^i = \nabla f_i(x_k)$ and

$$\left\| \nabla f_i(x_{k+1}) - v_k^i \right\| \overset{Equation\ 2}{\leq} L_{\max} \left\| x_{k+1} - x_k \right\|$$

$$= L_{\max} \gamma \left\| \nabla f(x_k) \right\|$$

$$\overset{Equation\ 15}{\leq} L_{\max} \gamma \left\| \nabla f(x_k) - v_{k-1} \right\| + L_{\max} \gamma \left\| v_{k-1} \right\|$$

$$\overset{Equation\ 15}{\leq} L_{\max} \gamma \frac{1}{n} \sum_{i=1}^{n} \left\| \nabla f_i(x_k) - v_{k-1}^i \right\| + L_{\max} \gamma \|v_{k-1}\|$$

$$\leq L_{\max} \gamma \tau + L_{\max} \gamma \left\| v_{k-1} \right\|.$$

From Claim I.3 and Equation 62,

$$\|v_{k-1}\| \leq \sqrt{\frac{4}{\gamma} \phi_0} + 2(B - \tau).$$

Hence, for all $i$

$$\left\| \nabla f_i(x_{k+1}) - v_k^i \right\| \leq L_{\max} \gamma \tau + 2 L_{\max} \sqrt{\gamma \phi_0} + 2 L_{\max} \gamma (B - \tau)$$

$$\leq 2 L_{\max} \sqrt{\gamma \phi_0} + 2 L_{\max} \gamma B \tag{63}$$

In conclusion, $\left\| \nabla f_i(x_{k+1}) - v_k^i \right\| \leq \tau$ is true for all $i$ if $\gamma$ satisfies

$$2 L_{\max} \gamma B \leq \frac{\tau}{2} \quad \text{and} \quad 2 L_{\max} \sqrt{\gamma \phi_0} \leq \frac{\tau}{2}. \tag{64}$$

From Claim I.4, the condition Equation 64 is hence satisfied when

$$\gamma \leq \frac{\tau}{4 B L_{\max}} \quad \text{and} \quad \gamma \leq \frac{\tau^2}{16 L_{\max}^2 (\sqrt{F_0} + \sqrt{F_0 + \frac{G_0 \tau}{2\sqrt{2\eta} L_{\max}}})^2},$$

where $\eta := \min \left\{ 1, \frac{\tau}{\max_i \|\nabla f_i(x_0)\|} \right\}$, $F_0 := f(x_0) - f_{\inf}$ and $G_0 := \sqrt{\frac{1}{n} \sum_{i=1}^{n} (\|\nabla f_i(x_0)\| - \tau)^2}$. Putting all the conditions on $\gamma$ together, we obtain the results.

## J  PROOF OF PROPOSITION 5.5

If $\|\nabla f_i(x_0)\| \leq \tau$, then from Equation 63 and step-size condition from Lemma 5.4, we prove that the clipping operator is always turned off for all $k \geq 0$, i.e., $\|\nabla f_i(x_k) - v_{k-1}^i\| \leq \tau$ implying that $v_k^i = \nabla f_i(x_k)$ for all $k \geq 0$.

If $\|\nabla f_i(x_0)\| > \tau$, then the clipping operator is turned on at the beginning. Moreover, for all $k \geq 0$ such that $\|\nabla f_i(x_k) - v_{k-1}^i\| > \tau$ (clipping is turned on), we have from the derivation of Equation 58 and the step-size condition of Lemma 5.4

$$\|\nabla f_i(x_k) - v_{k-1}^i\| \leq \|\nabla f_i(x_{k-1}) - v_{k-2}^i\| - \tau/2 \leq \ldots \leq \|\nabla f_i(x_0)\| - k\tau/2.$$

Therefore, the situation when $\tau < \|\nabla f_i(x_k) - v_{k-1}^i\|$ is possible only for $0 \leq k < k^\star$ with $k^\star = \frac{2}{\tau}(\|\nabla f_i(x_0)\| - \tau) + 1$. After that, the clipping operator always turns off, i.e., $\|\nabla f_i(x_k) - v_{k-1}^i\| \leq \tau$ for $k \geq k^\star$.

## K    PROOF OF THEOREM 5.6

Let $\hat{x}_K$ be selected uniformly at random from $\{x_0, x_1, \ldots, x_{K-1}\}$. Then,

$$\mathbf{E}\left[\|\nabla f(\hat{x}_K)\|^2\right] = \frac{1}{K} \sum_{k=0}^{K-1} \|\nabla f(x_k)\|^2.$$

From Lemma 5.4,

$$\mathbf{E}\left[\|\nabla f(\hat{x}_K)\|^2\right] \leq \frac{2}{\gamma} \frac{1}{K} \sum_{k=0}^{K-1} (\phi_k - \phi_{k+1}) = \frac{2(\phi_0 - \phi_K)}{\gamma K} \leq \frac{2\phi_0}{\gamma K}.$$

Next, consider the step-size condition:

$$\gamma \leq \min\left(\frac{\eta/L_{\max}}{8}, \frac{1}{L} \frac{1}{1 + \sqrt{1 + 2\beta_1}}, \frac{\tau^2}{16 L_{\max}^2 \left[\sqrt{F_0} + \sqrt{\beta_2}\right]^2}\right).$$

First, we have $\frac{1}{\eta} = \mathcal{O}(1 + \frac{\max_i \|\nabla f_i(x_0)\|}{\tau})$ and

$$\beta_1 \leq \frac{(1-\eta)^2(1 + 2/\eta)}{1 - (1-\eta)(1 - 2/\eta)} \leq \frac{1 + 2/\eta}{1 - (1-\eta)} = \frac{2}{\eta^2} + \frac{1}{\eta} = \mathcal{O}\left(\frac{1}{\eta^2}\right) = \mathcal{O}\left(1 + \frac{(\max_i \|\nabla f_i(x_0)\|)^2}{\tau^2}\right),$$

$$\beta_2 = F_0 + \frac{\tau G_0}{\sqrt{2\eta} L_{\max}} \leq F_0 + \frac{1}{\sqrt{2}} \frac{\tau \max(\tau, \max_i \|\nabla f_i(x_0)\|)}{\sqrt{\max\left(1, \frac{\tau}{\max_i \|\nabla f_i(x_0)\|}\right)} L_{\max}}$$

$$= F_0 + \frac{1}{\sqrt{2}} \frac{\tau \max_i \|\nabla f_i(x_0)\| \sqrt{\max\left(1, \frac{\tau}{\max_i \|\nabla f_i(x_0)\|}\right)}}{L_{\max}}$$

$$= \mathcal{O}\left(F_0 + \frac{\tau \max_i \|\nabla f_i(x_0)\|}{L_{\max}} + \frac{\tau^{3/2} \max_i \|\nabla f_i(x_0)\|^{1/2}}{L_{\max}}\right).$$

Using this, we estimate $1/\gamma$ as

$$\frac{1}{\gamma} = \max\left(\frac{8 L_{\max}}{\eta}, L(1 + \sqrt{1 + 2\beta_1}), \frac{16(L_{\max})^2 \left[\sqrt{F_0} + \sqrt{\beta_2}\right]^2}{\tau^2}\right)$$

$$= \mathcal{O}\left(\max(L, L_{\max})\left(1 + \frac{\max_i \|\nabla f_i(x_0)\|}{\tau}\right) + \frac{(L_{\max})^2 F_0}{\tau^2} + T\right)$$

$$= \mathcal{O}\left(\max(L, L_{\max})\left(1 + \sqrt{\tau \max_i \|\nabla f_i(x_0)\|}\right) + \frac{(L_{\max})^2 F_0}{\tau^2}\right),$$

where $T = \frac{L_{\max} \max_i \|\nabla f_i(x_0)\|}{\tau} + L_{\max} \sqrt{\tau \max_i \|\nabla f_i(x_0)\|}$. Therefore, since

$$\frac{A}{\gamma} = \frac{1}{2[1 - (1-\eta)(1 - \eta/2)]} = \mathcal{O}\left(\frac{1}{\eta}\right) = \mathcal{O}\left(1 + \frac{\max_i \|\nabla f_i(x_0)\|}{\tau}\right)$$

and

$$\frac{1}{n} \sum_{i=1}^{n} \|\nabla f_i(x_0) - v_{-1}^i\|^2 = \frac{1}{n} \sum_{i=1}^{n} \|\nabla f_i(x_0)\|^2 = \mathcal{O}(\max_i \|\nabla f_i(x_0)\|^2),$$

we have

$$\mathbf{E}\left[\|\nabla f(\hat{x}_K)\|^2\right] \leq \frac{2\phi_0}{\gamma K} = \frac{2\left(f(x_0) - f_{\inf} + A \frac{1}{n} \sum_{i=1}^{n} \|\nabla f_i(x_0) - v_{-1}^i\|^2\right)}{\gamma K}$$

$$= \mathcal{O}\left(\frac{\left(1 + \frac{\max_i \|\nabla f_i(x_0)\|}{\tau}\right) \max(F_0 \max(L_{\max}, L), \max_i \|\nabla f_i(x_0)\|^2) + \frac{(L_{\max})^2 (F_0)^2}{\tau^2}}{K}\right).$$

**Algorithm 3** Press-Clip21-GD (Error Feedback for Distributed Optimization with Compression and Clipping)

1: **Input:** initial iterate $x_0 \in \mathbb{R}^d$; learning rate $\gamma > 0$; initial gradient shifts $v_{-1}^1, \dots, v_{-1}^n \in \mathbb{R}^d$; clipping threshold $\tau > 0$; deterministic contractive compression $\mathcal{C} : \mathbb{R}^d \to \mathbb{R}^d$, i.e. $\|\mathcal{C}(v) - v\|^2 \leq (1 - \alpha)\|v\|^2$ for $0 < \alpha \leq 1$ and $v \in \mathbb{R}^d$
2: **for** $k = 0, 1, 2, \dots, K - 1$ **do**
3:     Broadcast $x_k$ to all workers
4:     **for** each worker $i = 1, \dots, n$ in parallel **do**
5:         Compute $g_k^i = \mathcal{C}(\text{clip}_\tau(\nabla f_i(x_k) - v_{k-1}^i))$
6:         Update $v_k^i = v_{k-1}^i + g_k^i$
7:     **end for**
8:     $v_k = v_{k-1} + \frac{1}{n}\sum_{i=1}^n g_k^i$
9:     $x_{k+1} = x_k - \gamma \frac{1}{n}\sum_{i=1}^n v_k^i$
10: **end for**

## L    Adding Communication Compression into the Mix

In this section, we consider Press-Clip21-GD, which is described in Algorithm 3. This algorithm attains the descent inequality as described in the next theorem:

**Theorem L.1.** *Consider the problem of minimizing $f(x) = \frac{1}{n}\sum_{i=1}^n f_i(x)$. Suppose that each $f_i(x)$ is L-Lipschitz gradient and the whole objective $f(x)$ is lower bounded by $f_{\inf}$. Let $v_{-1}^i = 0$ for all $i$, $\eta := \min\left(1, \frac{\tau}{\max_i \|\nabla f_i(x_0)\|}\right)$, $F_0 := f(x_0) - f_{\inf}$, and $G_0 := \sqrt{\frac{1}{n}\sum_{i=1}^n (\max\{0, \|\nabla f_i(x_0)\| - \tau\} + \sqrt{1 - \alpha}\tau)^2}$. Then, Press-Clip21-GD with*

$$\gamma \leq \min\left(\frac{(1 - \sqrt{1-\alpha})\eta}{8L_{\max}}, \frac{1 - 1/\sqrt{2}}{L(1 + \sqrt{1 + 2\beta_1})}, \frac{(1 - \sqrt{1-\alpha})^2\tau^2}{16L_{\max}^2\left[\sqrt{F_0} + \sqrt{\beta_2}\right]^2}\right), \tag{65}$$

*where $\beta_1 := 2\frac{\max\{(1-\beta)(1+2/\beta), (1-\alpha)(1+2/\alpha)\}}{\beta}\left(\frac{L_{\max}}{L}\right)^2$ and $\beta_2 := F_0 + \frac{G_0(1 - \sqrt{1-\alpha})\tau}{\sqrt{2\beta}L_{\max}}$ satisfies*

$$\phi_{k+1} \leq \phi_k - \frac{\gamma}{2}\|\nabla f(x_k)\|^2, \tag{66}$$

*where $\phi_k := f(x_k) - f_{\inf} + \frac{\gamma}{n\beta}\sum_{i=1}^n \|\nabla f_i(x_k) - v_k^i\|^2$, and $\beta := 1 - (B_1 + B_2(1 - \eta)^2) \in [0, \alpha]$ with $B_1 = (1 - \alpha) + (1 + 1/\theta_1)(1 + \theta_2)(1 - \alpha)$, $B_2 = (1 + \theta_1) + (1 + 1/\theta_1)(1 + 1/\theta_2)(1 - \alpha)$ for some $\theta_1, \theta_2 > 0$ and some $\alpha \in (0, 1]$. In addition,*

$$\mathbf{E}\left[\|\nabla f(\hat{x}_K)\|^2\right] \leq \frac{2}{\gamma}\frac{\phi_0}{K}, \tag{67}$$

*where $\hat{x}_K$ is the point selected uniformly at random from $\{x_0, x_1, \dots, x_K\}$ for $K \geq 1$.*

## M    Proof for Theorem L.1

Press-Clip21-GD can be equivalently expressed as:

$$x_{k+1} = x_k - \gamma v_k, \tag{68}$$

where

$$v_k = \frac{1}{n}\sum_{i=1}^n v_k^i, \tag{69}$$

$$v_k^i = (1 - \eta_k^i)v_{k-1}^i + \eta_k^i \nabla f_i(x_k) + e_k^i, \tag{70}$$

and $e_k^i = \mathcal{C}(\text{clip}_\tau(\nabla f_i(x_k) - v_{k-1}^i)) - \text{clip}_\tau(\nabla f_i(x_k) - v_{k-1}^i)$ and $\eta_k^i = \min\left(1, \frac{\tau}{\|\nabla f_i(x_k) - v_{k-1}^i\|}\right)$. Note that the compressor $\mathcal{C}$ is contractive with $\alpha \in (0, 1]$, i.e.

$$\|\mathcal{C}(v) - v\|^2 \leq (1 - \alpha)\|v\|^2, \quad \forall v \in \mathbb{R}^d. \tag{71}$$

Also recall that the Lyapunov function for this analysis is

$$\phi_k = f(x_k) - f_{\inf} + A\frac{1}{n}\sum_{i=1}^{n}\|\nabla f_i(x_k) - v_k^i\|^2, \tag{72}$$

where $A = \frac{\gamma}{\beta}$ and $\beta := 1-(B_1+B_2(1-\eta)^2) \in [0,\alpha]$ with $B_1 = (1-\alpha)+(1+1/\theta_1)(1+\theta_2)(1-\alpha)$, $B_2 = (1+\theta_1)+(1+1/\theta_1)(1+1/\theta_2)(1-\alpha)$ for some $\theta_1, \theta_2 > 0$ and some $\alpha \in (0,1]$.

## M.1 USEFUL CLAIMS

We begin by presenting claims which are useful for deriving the result.

**Claim M.1.** *Let each $f_i$ have $L_i$-Lipschitz gradient. Then, for $k \geq 0$,*

$$\left\|\nabla f_i(x_{k+1}) - v_k^i\right\| \leq \max\{0, \left\|\nabla f_i(x_k) - v_{k-1}^i\right\| - \tau\} + L_{\max}\gamma\|v_k\| + \sqrt{1-\alpha}\tau. \tag{73}$$

*Proof.* From the definition of $v_k^i$,

$$
\begin{aligned}
\left\|\nabla f_i(x_{k+1}) - v_k^i\right\| &\overset{Equation\ 15}{\leq} \left\|\nabla f_i(x_k) - v_k^i\right\| + \left\|\nabla f_i(x_{k+1}) - \nabla f_i(x_k)\right\| \\
&\overset{Equation\ 2+Equation\ 68}{\leq} \left\|\nabla f_i(x_k) - v_k^i\right\| + L_{\max}\gamma\|v_k\| \\
&\overset{Equation\ 70}{=} \left\|\nabla f_i(x_k) - v_{k-1}^i - \mathcal{C}(\mathsf{clip}_\tau(\nabla f_i(x_k) - v_{k-1}^i))\right\| + L_{\max}\gamma\|v_k\| \\
&\overset{Equation\ 15}{\leq} \left\|\nabla f_i(x_k) - v_{k-1}^i - \mathsf{clip}_\tau(\nabla f_i(x_k) - v_{k-1}^i)\right\| + L_{\max}\gamma\|v_k\| \\
&\qquad + \left\|\mathsf{clip}_\tau(\nabla f_i(x_k) - v_{k-1}^i) - \mathcal{C}(\mathsf{clip}_\tau(\nabla f_i(x_k) - v_{k-1}^i))\right\| \\
&\overset{(Lemma\ 4.1(ii)\text{-}(iii))}{\leq} \max\{0, \left\|\nabla f_i(x_k) - v_{k-1}^i\right\| - \tau\} + L_{\max}\gamma\|v_k\| \\
&\qquad + \left\|\mathsf{clip}_\tau(\nabla f_i(x_k) - v_{k-1}^i) - \mathcal{C}(\mathsf{clip}_\tau(\nabla f_i(x_k) - v_{k-1}^i))\right\| \\
&\overset{Equation\ 71}{\leq} \max\{0, \left\|\nabla f_i(x_k) - v_{k-1}^i\right\| - \tau\} + L_{\max}\gamma\|v_k\| \\
&\qquad + \sqrt{1-\alpha}\left\|\mathsf{clip}_\tau(\nabla f_i(x_k) - v_{k-1}^i)\right\| \\
&\leq \max\{0, \left\|\nabla f_i(x_k) - v_{k-1}^i\right\| - \tau\} + L_{\max}\gamma\|v_k\| + \sqrt{1-\alpha}\tau.
\end{aligned}
$$

Here, the last inequality comes from the fact that $\|\mathsf{clip}_\tau(v)\| \leq \tau$ for $v \in \mathbb{R}^d$. $\qquad\square$

**Claim M.2.** *Let $v_{-1}^i = 0$ for all $i$, $\max_i \|\nabla f_i(x_0)\| := B > \tau$ and $\gamma \leq 2/L$. Then,*

$$\|v_0\| \leq \sqrt{\frac{4}{\gamma}\phi_0} + 2(B - [1 - \sqrt{1-\alpha}]\tau). \tag{74}$$

*Proof.* By the fact that $v_0 = \frac{1}{n}\sum_{i=1}^{n} v_0^i = \frac{1}{n}\sum_{i=1}^{n} \mathcal{C}(\text{clip}_\tau(\nabla f_i(x_0)))$,

$$
\begin{aligned}
\|v_0\| \quad &\overset{\text{Equation 15}}{\leq} \quad \|\nabla f(x_0)\| + \left\|\frac{1}{n}\sum_{i=1}^{n} \mathcal{C}(\text{clip}_\tau(\nabla f_i(x_0))) - \nabla f(x_0)\right\| \\
&\overset{\text{Equation 15}}{\leq} \quad \|\nabla f(x_0)\| + \frac{1}{n}\sum_{i=1}^{n} \|\mathcal{C}(\text{clip}_\tau(\nabla f_i(x_0))) - \nabla f_i(x_0)\| \\
&\overset{\text{Equation 15}}{\leq} \quad \|\nabla f(x_0)\| + \frac{1}{n}\sum_{i=1}^{n}[\|\mathcal{C}(\text{clip}_\tau(\nabla f_i(x_0))) - \text{clip}_\tau(\nabla f_i(x_0))\| \\
&\qquad\qquad + \|\text{clip}_\tau(\nabla f_i(x_0)) - \nabla f_i(x_0)\|] \\
&\overset{\text{Equation 71}}{\leq} \quad \|\nabla f(x_0)\| + \frac{1}{n}\sum_{i=1}^{n}[\sqrt{1-\alpha}\,\|\text{clip}_\tau(\nabla f_i(x_0))\| + \|\text{clip}_\tau(\nabla f_i(x_0)) - \nabla f_i(x_0)\|] \\
&\overset{\text{(Lemma 4.1(ii)-(iii))}}{\leq} \quad \|\nabla f(x_0)\| + \frac{1}{n}\sum_{i=1}^{n}[\sqrt{1-\alpha}\,\|\text{clip}_\tau(\nabla f_i(x_0))\| + \max\{0, \|\nabla f_i(x_0)\| - \tau\}] \\
&\leq \quad \|\nabla f(x_0)\| + \sqrt{1-\alpha}\,\tau + \frac{1}{n}\sum_{i=1}^{n}\max\{0, \|\nabla f_i(x_0)\| - \tau\}.
\end{aligned}
$$

Here, the last inequality comes from the fact that $\|\text{clip}_\tau(v)\| \leq \tau$ for $v \in \mathbb{R}^d$.

If $\max_i \|\nabla f_i(x_0)\| := B > \tau$ and $\gamma \leq 2/L$, then

$$
\begin{aligned}
\|v_0\| \quad &\leq \quad \|\nabla f(x_0)\| + \sqrt{1-\alpha}\,\tau + (B - \tau) \\
&\overset{\text{Equation 18}}{\leq} \quad \sqrt{2L[f(x_0) - f_{\inf}]} + \sqrt{1-\alpha}\,\tau + (B - \tau) \\
&\leq \quad \sqrt{2L\phi_0} + \sqrt{1-\alpha}\,\tau + (B - \tau) \\
&\leq \quad \sqrt{\frac{4}{\gamma}\phi_0} + \sqrt{1-\alpha}\,\tau + (B - \tau) \\
&\leq \quad \sqrt{\frac{4}{\gamma}\phi_0} + 2(B - [1 - \sqrt{1-\alpha}]\tau).
\end{aligned}
$$

$\square$

**Claim M.3.** *Fix $k \geq 1$. Let $f$ have $L$-Lipschitz gradient. Also suppose that $\|\nabla f_i(x_k) - v_{k-1}^i\| \leq \max_i \|\nabla f_i(x_0)\| := B > \tau$, $\|v_{k-1}\| \leq \sqrt{\frac{4}{\gamma}\phi_0} + 2(B - [1 - \sqrt{1-\alpha}]\tau)$, $\|\nabla f(x_{k-1})\| \leq \sqrt{\frac{2}{\gamma}\phi_0}$, and $\gamma \leq \frac{1-1/\sqrt{2}}{L}$, then*

$$
\|v_k\| \leq \sqrt{\frac{4}{\gamma}\phi_0} + 2(B - [1 - \sqrt{1-\alpha}]\tau). \tag{75}
$$

*Proof.* By the fact that $\nabla f(x) = \frac{1}{n}\sum_{i=1}^{n}\nabla f_i(x)$ and from the definition of $v_k^i$,

$$\|v_k\| \overset{\overset{Equation\ 15}{\leq}}{} \|\nabla f(x_k)\| + \left\|\frac{1}{n}\sum_{i=1}^{n}\mathcal{C}(\mathsf{clip}_\tau(m_k^i)) - m_k^i\right\|$$

$$\overset{\overset{Equation\ 15}{\leq}}{} \|\nabla f(x_k)\| + \frac{1}{n}\sum_{i=1}^{n}\left\|\mathcal{C}(\mathsf{clip}_\tau(m_k^i)) - m_k^i\right\|$$

$$\overset{\overset{Equation\ 15}{\leq}}{} \|\nabla f(x_k)\| + \frac{1}{n}\sum_{i=1}^{n}[\left\|\mathcal{C}(\mathsf{clip}_\tau(m_k^i)) - \mathsf{clip}_\tau(m_k^i)\right\| + \left\|\mathsf{clip}_\tau(m_k^i) - m_k^i\right\|]$$

$$\overset{\overset{Equation\ 71}{\leq}}{} \|\nabla f(x_k)\| + \frac{1}{n}\sum_{i=1}^{n}[\sqrt{1-\alpha}\left\|\mathsf{clip}_\tau(m_k^i)\right\| + \left\|\mathsf{clip}_\tau(m_k^i) - m_k^i\right\|]$$

$$\overset{\overset{(Lemma\ 4.1(ii)\text{-}(iii))}{\leq}}{} \|\nabla f(x_k)\| + \frac{1}{n}\sum_{i=1}^{n}[\sqrt{1-\alpha}\left\|\mathsf{clip}_\tau(m_k^i)\right\| + \max\{0, \left\|m_k^i\right\| - \tau\}]$$

$$\leq \|\nabla f(x_k)\| + \sqrt{1-\alpha}\tau + \frac{1}{n}\sum_{i=1}^{n}\max\{0, \left\|m_k^i\right\| - \tau\},$$

where $m_k^i = \nabla f_i(x_k) - v_{k-1}^i$. The last inequality comes from the fact that $\|\mathsf{clip}_\tau(v)\| \leq \tau$ for $v \in \mathbb{R}^d$.

If $\left\|\nabla f_i(x_k) - v_{k-1}^i\right\| \leq \max_i \|\nabla f_i(x_0)\| := B > \tau$, then

$$\|v_k\| \leq \|\nabla f(x_k)\| + \sqrt{1-\alpha}\tau + (B-\tau)$$

$$\overset{\overset{Equation\ 15}{\leq}}{} \|\nabla f(x_{k-1})\| + \|\nabla f(x_k) - \nabla f(x_{k-1})\| + \sqrt{1-\alpha}\tau + (B-\tau)$$

$$\overset{\overset{Equation\ 3+Equation\ 68}{\leq}}{} \|\nabla f(x_{k-1})\| + L\gamma\|v_{k-1}\| + \sqrt{1-\alpha}\tau + (B-\tau).$$

If $\|v_{k-1}\| \leq \sqrt{\frac{4}{\gamma}\phi_0} + 2(B - [1-\sqrt{1-\alpha}]\tau)$ and $\|\nabla f(x_{k-1})\| \leq \sqrt{\frac{2}{\gamma}\phi_0}$, then

$$\|v_k\| \leq (L\gamma + 1/\sqrt{2})\sqrt{\frac{4}{\gamma}\phi_0} + (2L\gamma + 1)(B - [1-\sqrt{1-\alpha}]\tau).$$

If $\gamma \leq \frac{1-1/\sqrt{2}}{L}$, then $\gamma \leq 1/(2L)$ and

$$\|v_k\| \leq \sqrt{\frac{4}{\gamma}\phi_0} + 2(B - [1-\sqrt{1-\alpha}]\tau).$$

$\square$

**Claim M.4.** *If*

$$0 < \gamma \leq \frac{(1-\sqrt{1-\alpha})^2\tau^2}{16L_{\max}^2\left(\sqrt{F_0} + \sqrt{F_0 + \frac{G_0(1-\sqrt{1-\alpha})\tau}{2\beta L_{\max}}}\right)^2}, \tag{76}$$

*for $F_0 = f(x_0) - f_{\inf}$, $G_0 = \sqrt{\frac{1}{n}\sum_{i=1}^{n}(\max\{0, \|\nabla f_i(x_0)\| - \tau\} + \sqrt{1-\alpha}\tau)^2}$ and $\beta \geq 0$, then*

$$4L_{\max}\sqrt{\gamma\phi_0} \leq \frac{(1-\sqrt{1-\alpha})\tau}{2}. \tag{77}$$

*Proof.* By the definition of $\phi_0$,

$$4L_{\max}\sqrt{\gamma\phi_0} = 4L_{\max}\sqrt{\gamma F_0 + \frac{\gamma^2}{\beta}\tilde{G}_0}$$

$$\overset{\overset{Equation\ 15}{\leq}}{} 4L_{\max}\sqrt{\gamma}\sqrt{F_0} + 4L_{\max}\frac{\gamma}{\sqrt{\beta}}\sqrt{\tilde{G}_0},$$

where $F_0 = f(x_0) - f_{\inf}$ and $\tilde{G}_0 = \frac{1}{n} \sum_{i=1}^n \left\| \nabla f_i(x_0) - v_0^i \right\|^2$. Since $v_{-1}^i = 0$ for all $i$, we have $v_0^i = \mathcal{C}(\mathsf{clip}_\tau(\nabla f_i(x_0)))$ and

$$
\begin{aligned}
\left\| \nabla f_i(x_0) - v_0^i \right\| \quad & \overset{Equation\ 15}{\leq} \quad \left\| \nabla f_i(x_0) - \mathsf{clip}_\tau(\nabla f_i(x_0)) \right\| \\
& \qquad + \left\| \mathsf{clip}_\tau(\nabla f_i(x_0)) - \mathcal{C}(\mathsf{clip}_\tau(\nabla f_i(x_0))) \right\| \\
& \overset{Equation\ 71}{\leq} \quad \left\| \nabla f_i(x_0) - \mathsf{clip}_\tau(\nabla f_i(x_0)) \right\| + \sqrt{1-\alpha} \left\| \mathsf{clip}_\tau(\nabla f_i(x_0)) \right\| \\
& \leq \quad \left\| \nabla f_i(x_0) - \mathsf{clip}_\tau(\nabla f_i(x_0)) \right\| + \sqrt{1-\alpha}\tau \\
& \overset{(Lemma\ 4.1(ii)\text{-}(iii))}{\leq} \quad \max\{0, \|\nabla f_i(x_0)\| - \tau\} + \sqrt{1-\alpha}\tau.
\end{aligned}
$$

Therefore,

$$
4L_{\max}\sqrt{\gamma \phi_0} \quad \leq \quad 4L_{\max}\sqrt{\gamma}\sqrt{F_0} + 4L_{\max}\frac{\gamma}{\sqrt{\beta}}G_0,
$$

where $G_0 = \sqrt{\frac{1}{n}\sum_{i=1}^n (\max\{0, \|\nabla f_i(x_0)\| - \tau\} + \sqrt{1-\alpha}\tau)^2}$.

Hence, $\gamma > 0$ satisfying

$$
4L_{\max}\sqrt{\gamma}\sqrt{F_0} + 4L_{\max}\frac{\gamma}{\sqrt{\beta}}G_0 \leq \frac{(1-\sqrt{1-\alpha})\tau}{2} \tag{78}
$$

also satisfies $4L_{\max}\sqrt{\gamma \phi_0} \leq \frac{(1-\sqrt{1-\alpha})\tau}{2}$. This condition Equation 78 can be expressed equivalently as:

$$
\frac{1}{\sqrt{\gamma}} - \frac{8\sqrt{F_0}}{(1-\sqrt{1-\alpha})\tau}L_{\max} - \sqrt{\gamma}\frac{8G_0}{\sqrt{\beta}(1-\sqrt{1-\alpha})\tau L_{\max}}L_{\max}^2 \geq 0.
$$

Applying Lemma D.4 with $L = L_{\max}$, $\beta_1 = \frac{8\sqrt{F_0}}{(1-\sqrt{1-\alpha})\tau}$ and $\beta_2 = \frac{8G_0}{\sqrt{\beta}(1-\sqrt{1-\alpha})\tau L_{\max}}$ yields

$$
0 < \sqrt{\gamma} \leq \frac{(1-\sqrt{1-\alpha})\tau}{4L_{\max}\left(\sqrt{F_0} + \sqrt{F_0 + \frac{G_0(1-\sqrt{1-\alpha})\tau}{2\beta L_{\max}}}\right)}.
$$

Finally, taking the square, we obtain the final result. $\qquad\square$

## M.2 PROOF FOR EQUATION 66

To derive our result, let $\mathcal{I}_k$ be the subset from $\{1, 2, \ldots, n\}$ such that $\|\nabla f_i(x_k) - v_{k-1}^i\| > \tau$. We then derive the descent inequality .

$$
\phi_{k+1} \leq \phi_k - \frac{\gamma}{2}\|\nabla f(x_k)\|^2,
$$

for two possible cases: (1) when $|\mathcal{I}_k| > 0$ and (2) when $|\mathcal{I}_k| = 0$. For notational convenience we assume that $\nabla f_i(x_{-1}) = v_{-1}^i = 0$ and $\phi_{-1} = \phi_0$

### M.2.1 CASE (1): $|\mathcal{I}_k| > 0$

Suppose $|\mathcal{I}_k| > 0$. To derive the descent inequality we will show by induction the stronger result: for any $k \geq 0$, $\|\nabla f_i(x_k) - v_{k-1}^i\| \leq B - \frac{k\tau}{2}$ where $B := \max_i \|\nabla f_i(x_0)\|$ and

$$
\phi_k \leq \phi_{k-1} - \frac{\gamma}{2}\|\nabla f(x_{k-1})\|^2 - \frac{\gamma}{4}\|v_{k-1}\|^2. \tag{79}
$$

First when $k = 0$ we have $\|\nabla f_i(x_0) - v_{-1}^i\| = \|\nabla f_i(x_0)\| \leq \max_i \|\nabla f_i(x_0)\| = B$ for $B > \tau$ and for $i \in \mathcal{I}_{-1}$ and Equation 79 holds by definition. Next, we assume that $\|\nabla f_i(x_t) - v_{t-1}^i\| \leq B$ and Equation 79 hold for $t = 0, 1, \ldots, k$.

If $\gamma \leq (1 - 1/\sqrt{2})/L$, then $\gamma \leq 1/L$ and

$$
\begin{aligned}
\|\nabla f(x_{k-1})\|^2 \quad &\overset{Equation\ 18}{\leq} \quad 2L[f(x_{k-1}) - f_{\inf}] \\
&\overset{Equation\ 26}{\leq} \quad 2L\phi_{k-1} \\
&\leq \quad 2L\phi_0 \\
&\leq \quad \frac{2}{\gamma}\phi_0.
\end{aligned}
$$

By using the above inequality, and Claim M.2 and M.3, for $k \geq 0$

$$
\|v_k\| \leq \sqrt{\frac{4}{\gamma}\phi_0} + 2(B - [1 - \sqrt{1-\alpha}]\tau).
$$

Next, from the above inequality, and from Claim M.1,

$$
\begin{aligned}
\left\|\nabla f_i(x_{k+1}) - v_k^i\right\| \quad &\leq \quad \left\|\nabla f_i(x_k) - v_{k-1}^i\right\| - \tau + L_{\max}\gamma\|v_k\| + \sqrt{1-\alpha}\tau \\
&\leq \quad \left\|\nabla f_i(x_k) - v_{k-1}^i\right\| - [1 - \sqrt{1-\alpha}]\tau + L_{\max}\gamma\sqrt{\frac{4}{\gamma}\phi_0} \\
&\quad + 2L_{\max}\gamma(B - [1 - \sqrt{1-\alpha}]\tau) \\
&= \quad \left\|\nabla f_i(x_k) - v_{k-1}^i\right\| - [1 - \sqrt{1-\alpha}]\tau + 2L_{\max}\sqrt{\gamma\phi_0} \\
&\quad + 2L_{\max}\gamma(B - [1 - \sqrt{1-\alpha}]\tau).
\end{aligned}
$$

**STEP: Small step-size**

The above inequality and the inductive assumption imply:

$$
\|\nabla f_i(x_{k+1}) - v_k^i\| \leq \|\nabla f_i(x_k) - v_{k-1}^i\| - (1 - \sqrt{1-\alpha})\frac{\tau}{2} \leq B - \frac{(k+1)(1 - \sqrt{1-\alpha})\tau}{2} \tag{80}
$$

if the step-size $\gamma > 0$ satisfies

$$
2L_{\max}\gamma B \leq \frac{(1 - \sqrt{1-\alpha})\tau}{4} \quad \text{and} \quad 4L_{\max}\sqrt{\gamma\phi_0} \leq \frac{(1 - \sqrt{1-\alpha})\tau}{2}. \tag{81}
$$

By Claim M.4, this condition can rewritten into:

$$
\gamma \leq \frac{(1 - \sqrt{1-\alpha})\tau}{8BL_{\max}} \quad \text{and} \quad \gamma \leq \frac{(1 - \sqrt{1-\alpha})^2\tau^2}{16L_{\max}^2\left(\sqrt{F_0} + \sqrt{F_0 + \frac{G_0(1 - \sqrt{1-\alpha})\tau}{2\beta L_{\max}}}\right)^2}.
$$

**STEP: Descent inequality**

It remains to prove the descent inequality. By the inductive assumption proved above, We then have for $i \in \mathcal{I}_{k+1}$

$$
\eta_{k+1}^i = \frac{\tau}{\left\|\nabla f_i(x_{k+1}) - v_k^i\right\|} \geq \frac{\tau}{B} := \eta. \tag{82}
$$

Therefore,

$$
\begin{aligned}
\left\|\nabla f_i(x_{k+1}) - v_{k+1}^i\right\|^2 \quad &= \quad \left\|\nabla f_i(x_{k+1}) - v_k^i - \mathcal{C}(\nabla f_i(x_{k+1}) - v_k^i)\right\|^2 \mathbb{1}(i \in \mathcal{I}_{k+1}') \\
&\quad + \|\nabla f_i(x_{k+1}) - v_k^i - \mathcal{C}(\mathsf{clip}_\tau(\nabla f_i(x_{k+1}) - v_k^i))\|^2 \mathbb{1}(i \in \mathcal{I}_{k+1}) \\
\overset{Equation\ 17}{\leq} \quad & \left\|\nabla f_i(x_{k+1}) - v_k^i - \mathcal{C}(\nabla f_i(x_{k+1}) - v_k^i)\right\|^2 \mathbb{1}(i \in \mathcal{I}_{k+1}') \\
&\quad + (1+\theta_1)\|\nabla f_i(x_{k+1}) - v_k^i - \mathsf{clip}_\tau(\nabla f_i(x_{k+1}) - v_k^i)\|^2 \mathbb{1}(i \in \mathcal{I}_{k+1}) \\
&\quad + (1+1/\theta_1)\|\mathsf{clip}_\tau(\nabla f_i(x_{k+1}) - v_k^i) - \mathcal{C}(\mathsf{clip}_\tau(\nabla f_i(x_{k+1}) - v_k^i))\|^2 \mathbb{1}(i \in \mathcal{I}_{k+1}) \\
\overset{Equation\ 71}{\leq} \quad & (1-\alpha)\left\|\nabla f_i(x_{k+1}) - v_k^i\right\|^2 \mathbb{1}(i \in \mathcal{I}_{k+1}') \\
&\quad + (1+\theta_1)\|\nabla f_i(x_{k+1}) - v_k^i - \mathsf{clip}_\tau(\nabla f_i(x_{k+1}) - v_k^i)\|^2 \mathbb{1}(i \in \mathcal{I}_{k+1}) \\
&\quad + (1+1/\theta_1)(1-\alpha)\|\mathsf{clip}_\tau(\nabla f_i(x_{k+1}) - v_k^i)\|^2 \mathbb{1}(i \in \mathcal{I}_{k+1}) \\
\overset{Equation\ 17}{\leq} \quad & B_1\left\|\nabla f_i(x_{k+1}) - v_k^i\right\|^2 \\
&\quad + B_2\|\nabla f_i(x_{k+1}) - v_k^i - \mathsf{clip}_\tau(\nabla f_i(x_{k+1}) - v_k^i)\|^2 \mathbb{1}(i \in \mathcal{I}_{k+1}) \\
\leq \quad & B_1\left\|\nabla f_i(x_{k+1}) - v_k^i\right\|^2 \\
&\quad + B_2(1-\eta_{k+1}^i)^2\|\nabla f_i(x_{k+1}) - v_k^i\|^2 \mathbb{1}(i \in \mathcal{I}_{k+1}) \\
\overset{Equation\ 82}{\leq} \quad & (B_1 + B_2(1-\eta)^2)\left\|\nabla f_i(x_{k+1}) - v_k^i\right\|^2,
\end{aligned}
$$

for $B_1 = (1-\alpha) + (1+1/\theta_1)(1+\theta_2)(1-\alpha)$, $B_2 = (1+\theta_1) + (1+1/\theta_1)(1+1/\theta_2)(1-\alpha)$, and $\theta_1, \theta_2 > 0$.

Suppose that there exists $\theta_1, \theta_2 > 0$ such that $1 - \beta := (B_1 + B_2(1-\eta)^2)$ for $\beta \in [0, \alpha]$. Then,

$$
\begin{aligned}
\left\|\nabla f_i(x_{k+1}) - v_{k+1}^i\right\|^2 \quad &\leq \quad (1-\beta)\left\|\nabla f_i(x_{k+1}) - v_k^i\right\|^2 \\
\overset{Equation\ 17}{\leq} \quad & (1+\theta)(1-\beta)\left\|\nabla f_i(x_k) - v_k^i\right\|^2 \\
&\quad + (1+1/\theta)(1-\beta)\left\|\nabla f_i(x_{k+1}) - \nabla f_i(x_k)\right\|^2 \\
\overset{Equation\ 2}{\leq} \quad & (1+\theta)(1-\beta)\left\|\nabla f_i(x_k) - v_k^i\right\|^2 \\
&\quad + (1+1/\theta)(1-\beta)L_{\max}^2\left\|x_{k+1} - x_k\right\|^2,
\end{aligned}
$$

for $\theta > 0$.

If $\theta = \beta/2$, then

$$
\left\|\nabla f_i(x_{k+1}) - v_{k+1}^i\right\|^2 \leq (1-\beta/2)\left\|\nabla f_i(x_k) - v_k^i\right\|^2 + (1+2/\beta)(1-\beta)L_{\max}^2\left\|x_{k+1} - x_k\right\|^2. \quad (83)
$$

Hence, we can obtain the descent inequality. From Lemma D.3,

$$
\begin{aligned}
\phi_{k+1} \quad &= \quad f(x_{k+1}) - f_{\inf} + A\frac{1}{n}\sum_{i=1}^{n}\left\|\nabla f_i(x_{k+1}) - v_{k+1}^i\right\|^2 \\[6pt]
&\overset{\text{Equation 20}}{\leq} \quad f(x_k) - f_{\inf} - \frac{\gamma}{2}\left\|\nabla f(x_k)\right\|^2 - \left(\frac{1}{2\gamma} - \frac{L}{2}\right)\left\|x_{k+1} - x_k\right\|^2 \\[6pt]
&\quad + \frac{\gamma}{2}\frac{1}{n}\sum_{i=1}^{n}\left\|\nabla f_i(x_k) - v_k^i\right\|^2 + A\frac{1}{n}\sum_{i=1}^{n}\left\|\nabla f_i(x_{k+1}) - v_{k+1}^i\right\|^2 \\[6pt]
&= \quad \phi_k - \frac{\gamma}{2}\left\|\nabla f(x_k)\right\|^2 - \left(\frac{1}{2\gamma} - \frac{L}{2}\right)\left\|x_{k+1} - x_k\right\|^2 \\[6pt]
&\quad + \left(\frac{\gamma}{2} - A\right)\frac{1}{n}\sum_{i=1}^{n}\left\|\nabla f_i(x_k) - v_k^i\right\|^2 + A\frac{1}{n}\sum_{i=1}^{n}\left\|\nabla f_i(x_{k+1}) - v_{k+1}^i\right\|^2 \\[6pt]
&\overset{\text{Equation 83}}{\leq} \quad \phi_k - \frac{\gamma}{2}\left\|\nabla f(x_k)\right\|^2 - \left(\frac{1}{2\gamma} - \frac{L}{2} - A\left(1 + \frac{2}{\beta}\right)(1-\beta)L_{\max}^2\right)\left\|x_{k+1} - x_k\right\|^2 \\[6pt]
&\quad + \left(\frac{\gamma}{2} + A(1-\beta/2) - A\right)\frac{1}{n}\sum_{i=1}^{n}\left\|\nabla f_i(x_k) - v_k^i\right\|^2 .
\end{aligned}
$$

Since $A = \frac{\gamma}{\beta}$,

$$
\phi_{k+1} \leq \phi_k - \frac{\gamma}{2}\left\|\nabla f(x_k)\right\|^2 - \left(\frac{1}{2\gamma} - \frac{L}{2} - \frac{\gamma}{\beta}\left(1 + \frac{2}{\beta}\right)(1-\beta)L_{\max}^2\right)\left\|x_{k+1} - x_k\right\|^2 .
$$

If the step-size $\gamma > 0$ satisfies

$$
\gamma \leq \frac{1}{L\left(1 + \sqrt{1 + \frac{4}{\beta}(1 + 2/\beta)(1-\beta)\frac{L_{\max}^2}{L^2}}\right)},
$$

then from Lemma D.4 with $\beta_1 = 2$ and $\beta_2 = \frac{4}{\beta}(1 + 2/\beta)(1-\beta)\frac{L_{\max}^2}{L^2}$, this condition implies that $\frac{1}{2\gamma} - \frac{L}{2} - \frac{\gamma}{\beta}\left(1 + \frac{2}{\beta}\right)(1-\beta)L_{\max}^2 \geq \frac{1}{4\gamma}$ and that

$$
\begin{aligned}
\phi_{k+1} \quad &\leq \quad \phi_k - \frac{\gamma}{2}\left\|\nabla f(x_k)\right\|^2 - \frac{1}{4\gamma}\left\|x_{k+1} - x_k\right\|^2 \\[6pt]
&\overset{\text{Equation 68}}{=} \quad \phi_k - \frac{\gamma}{2}\left\|\nabla f(x_k)\right\|^2 - \frac{\gamma}{4}\left\|v_k\right\|^2 .
\end{aligned}
$$

This concludes the proof in the case (1).

### M.2.2 CASE (2): $|\mathcal{I}_k| = 0$

Suppose $|\mathcal{I}_k| = 0$. Then, we show by the induction that for any $k \geq 0$, $\left\|\nabla f_i(x_k) - v_{k-1}^i\right\| \leq \tau$ for all $i$ and

$$
\phi_k \leq \phi_{k-1} - \frac{\gamma}{2}\|\nabla f(x_{k-1})\|^2 - \frac{\gamma}{4}\|v_{k-1}\|^2. \tag{84}
$$

First, the base case is trivial, i.e. when $k = 0$, we have $\left\|\nabla f_i(x_0) - v_{-1}^i\right\| \leq \|\nabla f_i(x_0)\| = \bar{B} \leq \tau$ and Equation 84 holds by the definition ($\phi_0 = \phi_{-1}$). Next, we assume $\left\|\nabla f_i(x_t) - v_{t-1}^i\right\| \leq \tau$ for all $i$ and Equation 84 hold for $t = 0, 1, \ldots, k$. We complete the proof by proving these conditions hold for $k + 1$.

If $\gamma \leq (1 - 1/\sqrt{2})/L$, then $\gamma \leq 1/L$ and

$$
\begin{aligned}
\|\nabla f(x_k)\|^2 &\overset{\substack{Equation\ 18}}{\leq} 2L[f(x_k) - f(x^\star)] \\
&\overset{\substack{Equation\ 26}}{\leq} 2L\phi_k \\
&\overset{\substack{Equation\ 84}}{\leq} 2L\phi_0 \\
&\leq \frac{2}{\gamma}\phi_0.
\end{aligned}
$$

Hence, from Claim M.1,

$$
\begin{aligned}
\left\|\nabla f_i(x_{k+1}) - v_k^i\right\| &\leq L_{\max}\gamma\|v_k\| + \sqrt{1-\alpha}\tau \\
&= L_{\max}\gamma\left\|\frac{1}{n}\sum_{i=1}^n v_{k-1}^i + \mathcal{C}(\nabla f_i(x_k) - v_{k-1}^i)\right\| + \sqrt{1-\alpha}\tau \\
&\overset{\substack{Equation\ 15}}{\leq} L_{\max}\gamma\|\nabla f(x_k)\| + L_{\max}\gamma\frac{1}{n}\sum_{i=1}^n \left\|(\nabla f_i(x_k) - v_{k-1}^i) - \mathcal{C}(\nabla f_i(x_k) - v_{k-1}^i)\right\| \\
&\quad + \sqrt{1-\alpha}\tau \\
&\overset{\substack{Equation\ 71}}{\leq} L_{\max}\gamma\|\nabla f(x_k)\| + L_{\max}\gamma\sqrt{1-\alpha}\frac{1}{n}\sum_{i=1}^n \left\|\nabla f_i(x_k) - v_{k-1}^i\right\| + \sqrt{1-\alpha}\tau \\
&\leq L_{\max}\gamma\|\nabla f(x_k)\| + (L_{\max}\gamma + 1)\sqrt{1-\alpha}\tau \\
&\leq L_{\max}\gamma\sqrt{\frac{2}{\gamma}\phi_0} + (L_{\max}\gamma + 1)\sqrt{1-\alpha}\tau
\end{aligned}
$$

**STEP: Small step-size**

$\|\nabla f_i(x_{k+1}) - v_k^i\| \leq \tau$ holds if the step-size $\gamma > 0$ satisfies

$$
\gamma \leq \frac{1 - \sqrt{1-\alpha}}{2\sqrt{1-\alpha}L_{\max}} \quad \text{and} \quad 4L_{\max}\sqrt{\gamma\phi_0} \leq \frac{(1 - \sqrt{1-\alpha})\tau}{2}.
$$

Here, the second condition is fulfilled when it satisfies Equation 81.

**STEP: Descent inequality**

It remains to prove the descent inequality. Note that Press-Clip21-GD reduces to EF21 at step $k$. Thus,

$$
\begin{aligned}
\|\nabla f_i(x_{k+1}) - v_{k+1}^i\|^2 &= \|\nabla f_i(x_{k+1}) - v_k^i - \mathcal{C}(\nabla f_i(x_{k+1}) - v_k^i)\|^2 \\
&\leq (1-\alpha)\|\nabla f_i(x_{k+1}) - v_k^i\|^2 \\
&\leq (1-\alpha)(1+\theta)\|\nabla f_i(x_k) - v_k^i\|^2 + (1-\alpha)(1+1/\theta)\|\nabla f_i(x_k) - \nabla f_i(x_{k-1})\|^2.
\end{aligned}
$$

By letting $\theta = \alpha/2$ and by the smoothness of each $f_i(x)$,

$$
\|\nabla f_i(x_{k+1}) - v_{k+1}^i\|^2 \leq (1-\alpha/2)\|\nabla f_i(x_k) - v_k^i\|^2 + (1-\alpha)(1+2/\alpha)(L_{\max})^2\|x_k - x_{k-1}\|^2. \tag{85}
$$

From the definition of $\phi_k$ and Lemma D.3,

$$\phi_{k+1} \leq f(x_k) - f_{\inf} - \frac{\gamma}{2} \|\nabla f(x_k)\|^2 - \left(\frac{1}{2\gamma} - \frac{L}{2}\right) \|x_{k+1} - x_k\|^2$$

$$+ \frac{\gamma}{2} \frac{1}{n} \sum_{i=1}^{n} \|\nabla f(x_k) - v_k\|^2 + A \frac{1}{n} \sum_{i=1}^{n} \|\nabla f_i(x_{k+1}) - v_{k+1}^i\|^2$$

$$= \phi_k - \frac{\gamma}{2} \|\nabla f(x_k)\|^2 - \left(\frac{1}{2\gamma} - \frac{L}{2}\right) \|x_{k+1} - x_k\|^2$$

$$+ \left(\frac{\gamma}{2} - A\right) \frac{1}{n} \sum_{i=1}^{n} \|\nabla f(x_k) - v_k\|^2 + A \frac{1}{n} \sum_{i=1}^{n} \|\nabla f_i(x_{k+1}) - v_{k+1}^i\|^2$$

$$\overset{\text{Equation 85}}{\leq} \phi_k - \frac{\gamma}{2} \|\nabla f(x_k)\|^2 - \left(\frac{1}{2\gamma} - \frac{L}{2} - A(1-\alpha)(1+2/\alpha)(L_{\max})^2\right) \|x_{k+1} - x_k\|^2$$

$$+ \left(\frac{\gamma}{2} + A(1-\alpha/2) - A\right) \frac{1}{n} \sum_{i=1}^{n} \|\nabla f_i(x_k) - v_k^i\|^2 .$$

Since $A = \frac{\gamma}{\beta}$ with $\beta \in [0, \alpha]$, we get

$$\phi_{k+1} \leq \phi_k - \frac{\gamma}{2} \|\nabla f(x_k)\|^2 - \frac{1}{2\gamma} \left(1 - \gamma L - \gamma^2 \cdot \frac{2(1-\alpha)(1+2/\alpha)}{\beta}(L_{\max})^2\right) \|x_{k+1} - x_k\|^2 .$$

If the step-size $\gamma$ satisfies

$$0 < \gamma \leq \frac{1}{L\left(1 + \sqrt{1 + \frac{4(1-\alpha)(1+2/\alpha)}{\beta}\left(\frac{L_{\max}}{L}\right)^2}\right)},$$

then from Lemma D.4 with $L = 1$, $\beta_1 = 2L$ and $\beta_2 = \frac{4(1-\alpha)(1+2/\alpha)}{\beta}\left(\frac{L_{\max}}{L}\right)^2$, this condition implies $1 - \gamma L - \gamma^2 \cdot \frac{2(1-\alpha)(1+2/\alpha)}{\beta}(L_{\max})^2 \geq \frac{1}{2}$ and thus

$$\phi_{k+1} \leq \phi_k - \frac{\gamma}{2} \|\nabla f(x_k)\|^2 - \frac{1}{4\gamma} \|x_{k+1} - x_k\|^2 .$$

Since $x_{k+1} - x_k = -\gamma v_k$,

$$\phi_{k+1} \leq \phi_k - \frac{\gamma}{2} \|\nabla f(x_k)\|^2 - \frac{\gamma}{4} \|v_k\|^2 . \tag{86}$$

Putting all the conditions on $\gamma$ together, we obtain the results.

## M.3 PROOF FOR THE CONVERGENCE BOUND EQUATION 67

Let $\hat{x}_K$ be selected uniformly at random from $\{x_0, x_1, \ldots, x_{K-1}\}$. Then,

$$\mathbf{E}\left[\|\nabla f(\hat{x}_K)\|^2\right] = \frac{1}{K} \sum_{k=0}^{K-1} \|\nabla f(x_k)\|^2 .$$

From Equation 66,

$$\mathbf{E}\left[\|\nabla f(\hat{x}_K)\|^2\right] \leq \frac{2}{\gamma} \frac{1}{K} \sum_{k=0}^{K-1} (\phi_k - \phi_{k+1}) = \frac{2(\phi_0 - \phi_K)}{\gamma K} \leq \frac{2\phi_0}{\gamma K} .$$

