# OpenReview forum: "Clip21: Error Feedback for Gradient Clipping"
_ICLR.cc/2024/Conference — Submitted to ICLR 2024_

### Official Review · Reviewer_Sgc9 · 2023-10-27

**Soundness:** 3 good
**Presentation:** 3 good
**Contribution:** 2 fair
**Rating:** 5
**Confidence:** 3

**Summary:**

Gradient clipping is a commonly-used technique in training deep neural networks to resolve the exploding gradient issue. This paper shows that in a distributed setting, naively clipping the client gradient would result in large estimation error. To overcome this challenge, this paper proposes the Clip21 mechanism inspired by the error-feedback framework. The authors establish theoretical guarantee for the convergence rate of Clip21 to stationary points, and the rate is faster than previous works. Finally, experiments are conducted to demonstrate the efficiency of the proposed approach.

**Strengths:**

1) The Clip21 method proposed in this paper seems to be novel. Although there are previous works on the error-feedback framework, none considers the setting of clipped gradients.

2) The authors establish rigorous theoretical analysis for Clip21 under standard assumptions. Notably, heterogeneous client objective functions are allowed.

3) Overall speaking, the paper is well written and all definitions and theorem are stated clearly.

**Weaknesses:**

1) While gradient clipping is originally known to solve the exploding gradient problem, it seems not reflected in theoretical analysis of this paper. This paper still considers the standard $L$-smooth setting rather than $(L_0,L_1)$-smoothness, so the benefit of using clipped gradients is unclear from the theory.

2) This paper only considers the case for deterministic gradients. Although it is expected that similar results hold for stochastic gradients, in the gradient clipping literature more restrictive assumptions are made on noise (bounded noise tather than the more standard bounded variance, see e.g. [1]). It is unclear whether gradient noise is an issue in the distributed setting.

[1] Zhang, J., He, T., Sra, S., & Jadbabaie, A. (2019). Why gradient clipping accelerates training: A theoretical justification for adaptivity. arXiv preprint arXiv:1905.11881.

**Questions:**

1) In your Example 1.1, what if one chooses different clipping thresholds for different clients, or use a 'soft clipping' $\nabla f(x)/(\|\nabla f(x)\|+a)$? In these cases the clipped gradient does not seem to cancel out.

2) (related to weaknesses 1) What is the motivation of considering gradient clipping at the client level? Because in a distributed setting, one only wants to optimize the averaged objective function, can we just clipp the averaged gradient at the server to accelerate training? If you clip the server's gradient instead of the clients, then the problem demonstrated by Example 1.1 does not exist, and there may be a more straightforward approach.

I am willing to increase my rating if my concerns are properly addressed.

---

> ### Author Response · Authors · 2023-11-21
> **Response to Reviewer Sgc9 [Part 1/2]**
>
> > **While gradient clipping is originally known to solve the exploding gradient problem, it seems not reflected in theoretical analysis of this paper. This paper still considers the standard $L$-smooth setting rather than $(L_0,L_1)$-smoothness, so the benefit of using clipped gradients is unclear from the theory.**
>
> We agree that Clip-GD enjoys stronger convergence guarantees than GD for problems under the $(L_0, L_1)$ smoothness assumption, which implies the exploding gradient condition in deep neural network training [1]. However, existing results of gradient clipping algorithms are limited for the centralized case (one master and one client).
>
> Even under the $L$-smoothness assumption, we show that distributed clipping algorithms do not even converge towards the optimal solution (see Example 1). This motivates our investigation to modify standard clipping methods to attain strong convergence for distributed settings. We also prove theoretically and empirically that Clip21-GD outperforms these standard gradient clipping algorithms.
>
> Furthermore, we believe that our analysis is sufficient to prove the convergence under the $(L_0, L_1)$ smoothness. We prove this by finding the upper-bound for $\Vert \nabla^2 f(x_k) \Vert$ and our results apply as follows:
>
> 1. By the $(L_0, L_1)$ smoothness and Assumption 3 of [A1],
>
> $$
> \Vert \nabla^2 f(x_k) \Vert \leq L_0 + L_1 \Vert \nabla f(x_k) \Vert \leq L_0 + L_1 \sqrt{2L[f(x_k)-f_{\inf}]}.
> $$
>
> 2. We then have $\Vert \nabla^2 f(x_k) \Vert \leq L_0 + L_1 \sqrt{2L\phi_0}$ for all $k \geq 0$ by proving that $f(x_k) \leq f_{\inf}+\phi_0$ by the induction and by setting the step-size $\gamma$ sufficiently small.
>
> We also refer the reviewer to our general response, where we explain why the current analysis from our paper holds under $(L_0, L_1)$-smoothness.
>
> [A1]: Khaled, Ahmed, and Peter Richtárik. "Better theory for SGD in the nonconvex world." arXiv preprint arXiv:2002.03329 (2020).
>
>
> > **This paper only considers the case for deterministic gradients. Although it is expected that similar results hold for stochastic gradients, in the gradient clipping literature more restrictive assumptions are made on noise (bounded noise tather than the more standard bounded variance, see e.g. [1]). It is unclear whether gradient noise is an issue in the distributed setting.**
>
> Stochastic optimization algorithms suffer from instability for neural network training, which is in the presence of heavy-tailed stochastic gradient noises. This has been observed empirically for training neural network models by centralized stochastic gradient descent in Zhang et al. (2020b), and distributed methods in [M1]. Furthermore, deriving the result for Clip21-GD with stochastic gradients is non-trivial, because of (1) the dependency of the clipping threshold $\tau$ on the noise variance parameters, and (2) the required condition on the bounded norm of the input vector before clipping. This may be proved by the strong bounded noise condition, e.g. $\Vert g - \nabla f(x) \Vert^\alpha \leq \sigma^\alpha$ with probability 1.
>
> [M1] Yang, Haibo, Peiwen Qiu, and Jia Liu. "Taming Fat-Tailed (“Heavier-Tailed” with Potentially Infinite Variance) Noise in Federated Learning." Advances in Neural Information Processing Systems 35 (2022): 17017-17029.
>
>
> > **In your Example 1.1, what if one chooses different clipping thresholds for different clients, or use a 'soft clipping' $\nabla f(x)/(\vert \nabla f(x) \vert + a)$? In these cases the clipped gradient does not seem to cancel out.**
>
> By using different clipping thresholds for different clients or soft clipping, the clipped gradient may not cancel out in Example 1.1. However, the solution using the clipped gradient from both cases will be further away from the initial solution for simple toy examples.
> First, we prove the case for different clipping thresholds by letting in Example 1.1 $\beta = 10$, $\alpha=-1$, $x_0=1$, and the clipping thresholds at $0.1$ for $f_1(x)$ and $1$ for $f_2(x)$. Then,
>
> $$
> x_1 =  x_0 - \gamma ( clip_{0.1}(\nabla f_1(x)) + clip_{0.1}(\nabla f_2(x))) = 1 + 0.9 \gamma.
> $$
>
> $x_1$ is thus further away from the optimum $x^\star=0$, as the step-size $\gamma>0$.
> Second, we prove the case for soft clipping by considering the problem of minimizing $f(x)=f_1(x)+f_2(x)+f_3(x)$ for $x\in\mathbb{R}$ and $f_i(x)=(\alpha_i/2) x^2$ for $\alpha_i>0$ and $i=1,2,3$. If we choose $\alpha_1 = 10$, $\alpha_2=-4$, $\alpha_3=-4$, $x_0=1$ and $a=1$, then $f$ has Lipschitz continuous gradient with $L=2$, and also
>
> $$
> x_1 =  x_0 - \gamma ( softclip_a(\nabla f_1(x)) + softclip_a(\nabla f_2(x)) + softclip_a(\nabla f_3(x))  ) = 1 + \gamma \frac{38}{55}.
> $$
>
> $x_1$ is therefore more distant from the optimum $x^\star=0$ as $\gamma>0$.

---

> ### Author Response · Authors · 2023-11-21
> **Response to Reviewer Sgc9 [Part 2/2]**
>
> > **(Related to weaknesses 1) What is the motivation of considering gradient clipping at the client level? Because in a distributed setting, one only wants to optimize the averaged objective function, can we just clipp the averaged gradient at the server to accelerate training? If you clip the server's gradient instead of the clients, then the problem demonstrated by Example 1.1 does not exist, and there may be a more straightforward approach.**
>
> Implementing the client-side gradient algorithm is necessary for designing federated averaging algorithms that stabilize neural network training while saving communication costs, e.g. CELGC by Liu et al. (2022). Clip21-GD can be modified for these federated learning applications to accelerate the convergence of CELGC by Liu et al. (2022). This is because distributed gradient clipping algorithms can be viewed as a special case of CELGC by Liu et al. (2022) with the number of local steps $I=1$. Also note that the implementation of clipping the average gradient of clients can be viewed as Clip21-GD with the $n=1$ case supported by our theory.

---

### Official Review · Reviewer_y5AM · 2023-11-01

**Soundness:** 3 good
**Presentation:** 3 good
**Contribution:** 2 fair
**Rating:** 5
**Confidence:** 4

**Summary:**

This article presents Clip21-GD, a new  client-side gradient clipping algorithm designed for distributed training. The inspiration for Clip21-GD is drawn from the error feedback mechanism EF-21, employed to accelerate the convergence of gradient-compression distributed optimization algorithms. Notably, Clip21-GD achieves a convergence rate of $O(1/K)$, the same as standard distributed Gradient Descent (GD), and provides a more refined theoretical convergence analysis compared to EF-21, highlighting distinctions between clipping and compression operations.

**Strengths:**

-The theoretical convergence analysis for Clip21-GD is a strong point, as it reveals valuable insights. The theoretical $O(1/K) $ convergence rate surpasses that of the CE-FedAvg, signifying faster convergence.

**Weaknesses:**

- The article's motivation may need further clarity. While it claims to address the importance of deep neural network (DNN) training, the use of gradient descent (GD) in Clip-GD may appear unsuitable for DNNs. While Clip-SGD is briefly mentioned in the appendix for VGG 11 training, its theoretical convergence rate is not thoroughly explored in the main text, leaving room for ambiguity.

- The necessity of a client-side gradient clipping algorithm is not sufficiently explained. The implementation of clipping the average gradient of clients on the server side may seem more straightforward and effective in managing explosive gradients to prevent convergence issues.

**Questions:**

The article's disclosure that gradient clipping in Clip21-GD will not work after a specific number of steps ($K=O(1/\tau)$) raises questions about the algorithm's practicality and long-term effectiveness. The purpose and usability of Clip21-GD beyond this threshold remain unclear.

---

> ### Author Response · Authors · 2023-11-21
> **Response to Reviewer y5AM**
>
> > **The article's motivation may need further clarity. While it claims to address the importance of deep neural network (DNN) training, the use of gradient descent (GD) in Clip-GD may appear unsuitable for DNNs. While Clip-SGD is briefly mentioned in the appendix for VGG 11 training, its theoretical convergence rate is not thoroughly explored in the main text, leaving room for ambiguity.**
>
> We believe that Clip21-GD can be modified to obtain similar convergence guarantees when it uses stochastic gradients. This is because EF21 that inspires Clip21-GD attains the $\mathcal{O}(K^{-1})$ convergence rate when it uses stochastic gradients for nonconvex problems (see Theorem 3 in Appendix D.1 of [L1]). From this theorem, the residual error term of running EF21-SGD decreases when we increase the mini-batch size of stochastic gradients.
>
> However, deriving the result for Clip21-GD with stochastic gradients is non-trivial because of the required condition on the bounded norm of the input vector before clipping. This condition can be proved by the strong bounded noise condition, e.g. $\Vert g - \nabla f(x) \Vert\leq \sigma$ with probability 1 [L2].
>
> [L1] Fatkhullin, Ilyas, Igor Sokolov, Eduard Gorbunov, Zhize Li, and Peter Richtárik. "EF21 with bells & whistles: Practical algorithmic extensions of modern error feedback." arXiv preprint arXiv:2110.03294 (2021).
>
> [L2] Zhang, Bohang, Jikai Jin, Cong Fang, and Liwei Wang. "Improved analysis of clipping algorithms for non-convex optimization." Advances in Neural Information Processing Systems 33 (2020): 15511-15521.
>
>
> > **The necessity of a client-side gradient clipping algorithm is not sufficiently explained. The implementation of clipping the average gradient of clients on the server side may seem more straightforward and effective in managing explosive gradients to prevent convergence issues.**
>
> We believe that implementing the client-side gradient algorithm is necessary for designing federated algorithms that stabilize neural network training while saving communication costs, e.g. CELGC by Liu et al. (2022). Clip21-GD can be extended for these federated set-ups to further accelerate the convergence of CELGC by Liu et al. (2022). This is because distributed gradient clipping algorithms can be viewed as a special case of CELGC by Liu et al. (2022) with the number of local steps $I=1$. Also note that the implementation of clipping the average gradient of clients can be viewed as Clip21-GD with the $n=1$ case supported by our theory.
>
> > **The article's disclosure that gradient clipping in Clip21-GD will not work after a specific number of steps ( $K=\mathcal{O}(1/\tau)$) raises questions about the algorithm's practicality and long-term effectiveness. The purpose and usability of Clip21-GD beyond this threshold remain unclear.**
>
> Clip21-GD and Clip-GD become classical gradient descent when the clipping threshold $\tau$ is extremely large. In this situation, we cannot expect both gradient clipping methods to outperform classical gradient descent. Furthermore, from Figure 1 and 2, Clip21-GD empirically outperforms Clip-GD, especially when the clipping threshold $\tau$ becomes small. This is because Clip21-GD allows for the clipping to be inactive sooner than Clip-GD. This verifies Proposition 5.2 and 5.5, stating that the step before clipping becomes inactive $K=(1/\tau)(\| \nabla f(x_0) \| - \tau)$ is large as the clipping threshold $\tau$ is small and the initial solution $x_0$ satisfies $\| \nabla f(x_0)\| \gg \tau$.

---

> ### Comment · Reviewer_y5AM · 2023-11-22
>
> I appreciate the responses from the author, yet my concerns persist.
>
> 1. The extension from Clip21-GD to Clip21-SGD is non-trivial, challenging the claim that Clip21-GD effectively addresses issues in DNN training. This motivation seems overstated.
>
> 2. The rationale behind implementing client-side gradient clipping remains unclear. Employing simpler server-side gradient clipping could also potentially achieve training stability, and the benefits of client-side clipping in reducing communication overhead are not clearly demonstrated.

---

> > ### Author Response · Authors · 2023-11-23
> > **Response to the comment by Reviewer y5AM**
> >
> > We thank the reviewer for engaging in the discussion with us.
> >
> > 1. We agree that the extension of Clip21-GD to Clip21-SGD is non-trivial under bounded variance assumption (since in our analysis, we bound the norm of the input vector by induction), but it can be done under an additional assumption that the noise is bounded with probability, which is a standard assumption in the literature on clipping [1]. We also emphasize that the study of deterministic case, which our paper does, is an important building part in building a strong theory of distributed methods with clipping.
> >
> > 2. We also point out that in the context of Differential Privacy (DP), it is important to apply gradient clipping on the client side (Abadi et al, 2016). Although we do not derive DP guarantees in our paper, this is an important direction for the future extension of our work.
> >
> > [1] Zhang, Bohang, Jikai Jin, Cong Fang, and Liwei Wang. "Improved analysis of clipping algorithms for non-convex optimization." Advances in Neural Information Processing Systems 33 (2020): 15511-15521.

---

### Official Review · Reviewer_HYZq · 2023-11-03

**Soundness:** 3 good
**Presentation:** 4 excellent
**Contribution:** 3 good
**Rating:** 6
**Confidence:** 3

**Summary:**

In this paper the authors study the convergence of Gradient Descent (GD) for the optimization of sum of functions $f:=\frac{1}{n}\sum_{i=1}^if_i(x_i)$ (which arises naturally in Empirical Risk Minimization) when each of the individual gradient $\nabla_xf_i(x)$ is clipped. They assume Lipschitz gradient. Their contribution are the following:
* naive gradient clipping fail convergence in simple cases, even on mean-estimation tasks
* by fixing the mean estimation task with clipped gradients, it is possible to fix GD+clipping algorithm itself
* different variants are proposed, with their own rates

The results are illustrated on logistic regression, *with* and *without* convex regularization (i.e. with a convex and a non-convex task). Importantly, the work generalizes the n=1 case, and retrieve the same rates as existing literature for n=1.

**Strengths:**

### Clarity
The paper is extremely well written. The position of this work to the related work is clear. The problem is clearly stated. There are very convincing examples, progressive explanations of the proof strategies, and the "right level" of details.

### Significance & Quality
Elementwise Gradient clipping is widespread in the Differential Privacy literature; therefore these theoretical results are significant. The hypothesis on $f$ are realistic even in the context of deep learning, making the paper relevant for the community.

**Weaknesses:**

## Practical relevance

While the paper is well written and provide interesting insights, I have a few concerns about its usefulness *in practice*.

### Relevance to Differential Privacy

Element-wise gradient clipping is typically used DP-SGD (Abadi et al, 2016). DP-SGD uses clipping to control the sensitivity of gradient steps. Then DP-SGD adds a Gaussian noise to the clipped gradient to create a Gaussian mechanism. In Alg 1, the clipped vectors are aggregated, but the influence of a noise $\zeta_i$ on the mean estimate $\frac{1}{n}\sum_iv_i$ is not studied, therefore it is not clear if this algorithm would work in the context of differential privacy.

### Relevance for exploding gradients

As written in the paper:

> we are not employing clipping as a tool for taming the exploding gradients problem, and our work is fully complementary to this literature. [...] Clipping after averaging does not cause the severe bias and divergence issues we are addressing in our work.

## Comparison against vanilla GD

The paper positions itself against the Clip-GD algorithm (among others... ), in all experiments. But experiments with comparison against **vanilla GD** are lacking.

### Conclusion

See my question below.

**Questions:**

### When do we care about the average ?

This is a high level question.

The idea of Algorithm 1 is to estimate the average $\frac{1}{n}\sum_ia_i$ *exactly*. Clipping may help to circumvent instabilities when some individual gradients $a_i$ are too high (whetever the reason, exploding gradient being an example, numerical inaccuracies another). In this case, **biasing the average direction** with clipping *is* the wanted behavior, since we don't want the descent step to be driven by a single example with exceedingly large gradient norm. The average operator $\bar a:=a\mapsto \frac{1}{n}\sum_ia_i$ is not robust by definition against outliers.

By building an estimator of this average operator $\bar a$, aren't we falling back to the issues the average operator $\bar a$ was suffering in the first place?

### Comparison against vanilla GD

Can you clarify what are the use-cases in which Clip21-GD overcomes **vanilla GD** (and not only Clip-GD) ? What makes your algorithm different from vanilla GD in practice? Experiments are lacking to compare against it.

### Typo
For citations, for example a top of page 5, I recommand to use `\citep{}` instead of `\cite{}`, for example in `Richt ´arik et al. (2021); Fatkhullin et al. (2021); Richt ´arik et al. (2022),`.

> it will become “inactive” in at most k⋆ steps (i.e., ∇fi(xk) − vik−1 ≤ τ

Missing `)`.

---

> ### Author Response · Authors · 2023-11-21
> **Response to Reviewer HYZq**
>
> > **Relevance to Differential Privacy. Element-wise gradient clipping is typically used DP-SGD (Abadi et al, 2016). DP-SGD uses clipping to control the sensitivity of gradient steps. Then DP-SGD adds a Gaussian noise to the clipped gradient to create a Gaussian mechanism. In Alg 1, the clipped vectors are aggregated, but the influence of a noise $\xi_i$  on the mean estimate $\frac{1}{n}\sum_i v_i$ is not studied, therefore it is not clear if this algorithm would work in the context of differential privacy.**
>
> Although DP-Clip-SGD for distributed settings (Abadi et al, 2016) has been studied empirically, there is no theoretical convergence for this method. Even without DP noise, distributed Clip-GD does not converge towards the optimum for quadratic problems in Example 1.1. This motivates us to redesign distributed Clip-GD that leverages the error-feedback framework to ensure convergence. Deriving DP guarantees for a version of Clip21-GD is a prominent research direction for future work.
>
> > **Relevance for exploding gradients. As written in the paper: ``we are not employing clipping as a tool for taming the exploding gradients problem, and our work is fully complementary to this literature. [...] Clipping after averaging does not cause the severe bias and divergence issues we are addressing in our work.”**
>
> We agree that clipping helps stabilize neural network training which has exploding gradient problems, which can be modeled by the $(L_0,L_1)$ smoothness condition. We believe that our analysis is sufficient to prove the convergence under the $(L_0, L_1)$ smoothness. We prove this by finding the upper-bound for $\Vert \nabla^2 f(x_k) \Vert$ and our results apply as follows:
>
> 1. By the $(L_0, L_1)$ smoothness and Assumption 3 of [A1],
>
> $$
> \Vert \nabla^2 f(x_k) \Vert \leq L_0 + L_1 \Vert \nabla f(x_k) \Vert \leq L_0 + L_1 \sqrt{2L[f(x_k)-f_{\inf}]}.
> $$
>
> 2. We then have $\Vert \nabla^2 f(x_k) \Vert \leq L_0 + L_1 \sqrt{2L\phi_0}$ for all $k \geq 0$ by proving that $f(x_k) \leq f_{\inf}+\phi_0$ by the induction and by setting the step-size $\gamma$ sufficiently small.
> We also refer the reviewer to our general response, where we explain why the current analysis from our paper holds under $(L_0, L_1)$-smoothness.
>
> [A1]: Khaled, Ahmed, and Peter Richtárik. "Better theory for SGD in the nonconvex world." arXiv preprint arXiv:2002.03329 (2020).
>
>
> > **The idea of Algorithm 1 is to estimate the average $\frac{1}{n}\sum_{i=1}a_i$ \textit{exactly}. Clipping may help to circumvent instabilities when some individual gradients $a_i$ are too high (whetever the reason, exploding gradient being an example, numerical inaccuracies another). In this case, biasing the average direction with clipping is the wanted behavior, since we don't want the descent step to be driven by a single example with exceedingly large gradient norm. The average operator $\bar a := a \mapsto \frac{1}{n}\sum_{i=1}a_i$ is not robust by definition against outliers.**
>
> > **By building an estimator of this average operator $\bar a$, aren't we falling back to the issues the average operator $\frac{1}{n}\sum_{i=1}a_i$ was suffering in the first place?**
>
> We agree that this naive averaging operator does not ensure the convergence of distributed clipped gradient methods as in Example 1.1. To this end, we show that this naive averaging can be circumvented by leveraging the error-feedback framework to recover the average of full gradients from the average of memory vectors. In particular, from Lemma 4.2 and 4.3, the Clip21 algorithm ensures the following: (1) each memory vector $v_k^i$ approaches $a^i$ at node $i$, and hence (2) we can obtain $v_k := \frac{1}{n}\sum_{i} v_i^k$ which approaches $a:= \frac{1}{n}\sum_{i} a_i$.
>
>
> > **The paper positions itself against the Clip-GD algorithm (among others... ), in all experiments. But experiments with comparison against vanilla GD are lacking.**
>
> > **Can you clarify what are the use-cases in which Clip21-GD overcomes vanilla GD (and not only Clip-GD) ? What makes your algorithm different from vanilla GD in practice? Experiments are lacking to compare against it.**
>
> Our theory and experiments suggest that Clip21-GD outperforms the Clip-GD for both deterministic and stochastic gradients since it can handle data heterogeneity and ensures strong convergence guarantees. We believe that Clip21-GD would outperform the naive clipping algorithm, which is faster than vanilla SGD for neural network trainin, e.g., in (Zhang et al.,2020a).
>
>
> > **For citations, for example a top of page 5, I recommand to use `\citep{}` instead of `\cite{}`, for example in, `Richt ´arik et al. (2021); Fatkhullin et al. (2021); Richt ´arik et al. (2022),`**
>
> > **it will become “inactive” in at most k⋆ steps (i.e., ∇fi(xk) − vik−1 ≤ τ. Missing $)$.**
>
> We thank the reviewer for this recommendation and for spotting this typo. We will correct this in our finalized manuscript.

---

### Author Response · Authors · 2023-11-21
**General response to the reviewers**

We thank the reviewers for their positive evaluations. All the reviewers appreciate the novelty and solid contributions of our proposed Clip-21 methods. In particular,

- Reviewer HYZq states that our ``theoretical results are significant. The hypothesis on $f$ are realistic even in the context of deep learning, making the paper relevant for the community.”

- Reviewer y5AM mentions that Our ``theoretical convergence analysis for Clip21-GD is a strong point, as it reveals valuable insights. The theoretical $\mathcal{O}(1/K)$ convergence rate surpasses that of the CE-FedAvg, signifying faster convergence.”

- Reviewer Sg9c says that the Clip21 method in this paper contains novelty, i.e. applying the error-feedback framework on the settings of clipped gradients, and we establish   ``rigorous theoretical analysis for Clip21 under standard assumptions. Notably, heterogeneous client objective functions are allowed.”

We also would like to address three major concerns among the reviewers as follows:

**Relevance for differential privacy and/or exploding gradient problems by $(L_0,L_1)$-smoothness.** Clipping plays an important role in differential privacy and in stabilizing neural network training with exploding gradient problems, which can be partially modeled by the $(L_0,L_1)$-smoothness condition. However, it is worth mentioning that $(L_0, L_1)$-smoothness implies $L$-smoothness on the compact: indeed, since the gradient is continuous mapping, its norm is bounded on every compact. This means that for any compact $X \subseteq \mathbb{R}^d$ there exits $G_X$ such that $\|\|\nabla f(x)\|\| \leq G_X$ for all $x \in X$, implying that $\|\| \nabla^2 f(x) \|\| \leq L_0 + L_1 \|\| \nabla f(x) \|\| \leq L_0 + L_1 G_X$ for all $x \in X$, i.e., $f$ is $L$-smooth on $X$ with $L = L_0 + L_1 G_X$. In our analysis, we show that all iterates $x_k$ generated by Clip21-GD satisfy $f(x_k) - f_{\inf} \leq \phi_0$, i.e., the function value remains bounded. Therefore, if the function is such that its level set $X_0 = \lbrace x \in \mathbb{R}^2\mid f(x) \leq f_{\inf} + \phi_0 \rbrace$ is bounded, which is a very mild assumption, then our analysis works under $(L_0,L_1)$-smoothness as well, since in this case $L$-smoothness holds with $L = L_0 + L_1 G_{X_0}$. However, this constant can be large, and thus, we agree that extending our analysis to the case of $(L_0,L_1)$-smoothness is a prominent research direction.

Regarding the differential privacy, we expect that our method should work in practice since in our experiments, Clip21-SGD (a stochastic version of Clip21-GD) performs relatively well compared to Clip-SGD, which is known to be a standard option for achieving DP. We leave this extension of our method and theory to future work.

Finally, although the mentioned extensions are important, we also point out that even a simpler problem – non-convergence of distributed Clip-GD under $L$-smoothness with fixed stepsize – was not addressed in the literature. In our work, we have resolved this issue via Clip21-GD. We believe it is an important step towards building more complicated and efficient distributed methods with clipping.

**The motivation of considering gradient clipping at the client level.** Implementing the client-level gradient algorithm is necessary for designing federated algorithms that stabilize neural network training while saving communication costs, e.g. CELGC by Liu et al. (2022). Clip21-GD can be extended for these federated set-ups to further accelerate the convergence of CELGC by Liu et al. (2022). This is because distributed gradient clipping algorithms can be viewed as a special case of CELGC by Liu et al. (2022) with the number of local steps $I=1$.

**Lack of the extension to stochastic gradients.** Our experiments also suggest that Clip21-SGD outperforms Clip-SGD. We believe that Clip21-GD in stochastic settings can be modified to obtain convergence guarantees similarly. This is because EF21 that inspires Clip21-GD attains the $\mathcal{O}(K^{-1})$ convergence rate when it uses stochastic gradients for nonconvex problems (see Theorem 3 in Appendix D.1 of [L1]). From this theorem, the residual error term of running EF21-SGD decreases when we increase the mini-batch size of stochastic gradients. However, deriving the result for Clip21-GD with stochastic gradients is non-trivial because of the required condition *in the current analysis* on the bounded norm of the input vector before clipping. This condition can be proved by the strong bounded noise condition, e.g. $\Vert g - \nabla f(x) \Vert\leq \sigma$ with probability 1 [L2].

To this end, we addressed every concern/question/comment of every reviewer individually in detail and kindly ask the reviewers to reconsider their scores based on our responses.

---

> ### Author Response · Authors · 2023-11-21
> **References**
>
> [L1] Fatkhullin, Ilyas, Igor Sokolov, Eduard Gorbunov, Zhize Li, and Peter Richtárik. "EF21 with bells & whistles: Practical algorithmic extensions of modern error feedback." arXiv preprint arXiv:2110.03294 (2021).
>
> [L2] Zhang, Bohang, Jikai Jin, Cong Fang, and Liwei Wang. "Improved analysis of clipping algorithms for non-convex optimization." Advances in Neural Information Processing Systems 33 (2020): 15511-15521.

---

### Meta-Review · Area_Chair_6T3g · 2023-12-09

**Metareview:**

- Inspired by EF21, the submission proposed Clip21 to reduce the bias induced by the gradient clipping operator and applied Clip21 to the GD algorithm for distributed training. The authors provide the convergence guarantee for Clip21, and the convergence rate is faster than previous works, which is the main contribution of this submission.
- The submission still has some flaws:
  - As also pointed out by several reviewers, the client-side gradient clipping algorithm is not common and not sufficiently explained. The authors' response does not address the concern, and the claim that the client-side gradient clipping could save communication costs makes things more confusing.
  - The motivation of the submission may need to be polished further. It aims to solve the problems in deep learning, however, the experimental setting is trivial. More importantly, the client-server communication mode was abandoned by the deep learning community several years ago. The author should pay attention to the expression of algorithms to make them closer to the current paradigm of deep learning if the author still wishes to use distributed training in deep learning as the motivation.
  - There exists a gap between Clip21-GD and Clip21-SGD, the latter is more important in practice. The discussion or the application for adopting the Clip21-GD method in federated learning is missing, which is quite important for explaining the reason to perform client-side gradient clipping behavior.

**Justification For Why Not Higher Score:**

Overall, the paper has made some theoretical contributions, but this contribution cannot compensate for the shortcomings in other aspects.

**Justification For Why Not Lower Score:**

N/A

---

### Decision · Program_Chairs · 2024-01-16

Reject